# HIGH PRECISION SCORE-BASED DIFFUSION MODELS

## ABSTRACT

Recent advances in diffusion models bring the state-of-the art performance on image generation tasks. However, image generation is still an arduous task in high resolution, both theoretically and practically. From the theory side, the difficulty arises in estimating the high precision diffusion because the data score goes to $\infty$ as $t \rightarrow 0$ of the diffusion time. This paper resolves this difficulty by improving the previous diffusion models from three aspects. First, we propose an alternative parameterization for such an unbounded data score, which theoretically enables the unbounded score estimation. Second, we provide a practical Soft Truncation trick (ST-trick) to handle the extreme variation of the score scales. Third, we design a Reciprocal Variance Exploding Stochastic Differential Equation (RVESDE) to enable the sampling at the high precision of $t$. These three improvements are applicable to the variations of both NCSN and DDPM, and our improved versions are named as HNCSN and HDDPM, respectively. The experiments show that the improvements result in the state-of-the-art performances in the high resolution image generation, i.e. CelebA-HQ. Also, our ablation study empirically illustrates that all of 1) alternative parameterization, 2) ST-trick, and 3) RVESDE contributes to the performance enhancement.

## 1 INTRODUCTION

Recent advances in the generative models enable creating of highly realistic images. One direction of such modeling is *likelihood-free models* (Karras et al., 2019) based on the minimax training. The other direction is *likelihood-based models*, including VAE Vahdat and Kautz (2020), autoregressive models (Parmar et al., 2018), and flow models (Grcić et al., 2021). Diffusion models (Ho et al., 2020) are one of the most successful *likelihood-based models* where the generative process is modeled by the reverse diffusion process. The success of diffusion models achieves the state-of-the-art performance in image generation (Dhariwal and Nichol, 2021; Song et al., 2020).

Despite of such success, the score-based diffusion models still struggle in generating realistic images with high resolution (Jolicoeur-Martineau et al., 2021). Rather than improving the network architecture, this paper focuses mostly on the theoretic and the optimization aspects of the diffusion model to improve the sample quality. We first observe that the diffusion time acts in distinctive ways on the sample generation: the diffusion process with the front diffusion time ($t \approx 0$) influences the local sample fidelity, and the end diffusion time ($t \approx T$) shapes the sample global structure. However, the unbounded data score at the front diffusion time results in the training instability, and this instability yields the diffusion model with poor estimation limiting the local sample fidelity, which prevents the advancement in the high resolution because of its locality. Hence, this paper constructs a diffusion model capable of training both front time and end time, effectively.

We observe that the current score parametrization is incapable of estimating the data score that blows up as $t \rightarrow 0$, which requires a new theory-driven parametrization for the unbounded score network to enable the successful score estimation near $t \approx 0$. This parametrization is universally applicable for any diffusion models including NCSN (Song and Ermon, 2019) and DDPM (Ho et al., 2020), and we call this parametrization as High precision NCSN (HNCSN) and High precision DDPM (HDDPM), respectively. Second, we propose a practical optimization method (ST-trick) to improve the score estimation at the end of the diffusion time ($t \approx T$). This ST-trick provides a distinctive way to optimize the diffusion model parameters, and we argue that this trick successfully trains the score network at the end diffusion time by its nature, which is likely to be ignored due to the minor contribution to the diffusion loss near $t \approx T$ in the previous studies. Like the new parametrization,

ST-trick fits to any diffusion model, including NCSN/HNCSN and DDPM/HDDPM. Finally, we suggest a new SDE (RVESDE) that 1) mitigates the theoretic dilemma of VESDE that arises as $t \to 0$ and 2) gives additional performance boost by reducing the Monte-Carlo variance for the diffusion loss. We perform the ablation study on our three improvements, and we find that using 1) the new parametrization, 2) ST-trick, and 3) RVESDE all together improves both density estimation and sample fidelity on benchmark datasets. Although we proceed our arguments based on the continuous diffusion models, we emphasize that both new parametrization and ST-trick are applicable for discrete diffusion models, such as ADM (Dhariwal and Nichol, 2021) and CDM (Ho et al., 2021), as well.

## 2 PRELIMINARY

The diffusion model trains the network by minimizing the Negative Evidence Lower BOund (NELBO)

$$\text{(NELBO)} \qquad \mathbb{E}_{p_r(\mathbf{x}_0)}\big[-\log p_{\boldsymbol{\theta}}(\mathbf{x}_0)\big] \leq \mathcal{L}(\boldsymbol{\theta}; \lambda = g^2) = \frac{1}{2}\int_0^T g^2(t)\mathcal{L}_t(\boldsymbol{\theta})\,\mathrm{d}t, \qquad (1)$$

where

$$\mathcal{L}_t(\boldsymbol{\theta}) = \mathbb{E}_{\mathbf{x}_t}\big[\|\mathbf{s}_{\boldsymbol{\theta}}(\mathbf{x}_t, t) - \nabla_{\mathbf{x}_t}\log p_t(\mathbf{x}_t)\|_2^2\big] = \mathbb{E}_{\mathbf{x}_0, \mathbf{x}_t}\big[\|\mathbf{s}_{\boldsymbol{\theta}}(\mathbf{x}_t, t) - \nabla_{\mathbf{x}_t}\log p_{0t}(\mathbf{x}_t|\mathbf{x}_0)\|_2^2\big],$$

up to a constant. Here, $\lambda$ is the weighting function for the diffusion loss of $\mathcal{L}_t(\boldsymbol{\theta})$ at each diffusion time, $p_t(\mathbf{x}_t)$ is the marginal distribution of the diffusion process of $\{\mathbf{x}_t\}_{t=0}^T$, and $p_{0t}(\mathbf{x}_t|\mathbf{x}_0)$ is the transition probability defined by $\mathcal{N}(\mathbf{x}_t; \mu(t)\mathbf{x}_0, \sigma^2(t)\mathbf{I})$, where $\mu(t)$ and $\sigma^2(t)$ are determined by the governing diffusion process. Previous works suggest a couple of diffusion processes: 1) VESDE $\mathrm{d}\mathbf{x}_t = g(t)\,\mathrm{d}\boldsymbol{\omega}_t$ with $g(t) = \sigma_{min}(\frac{\sigma_{max}}{\sigma_{min}})^t\sqrt{2\log(\frac{\sigma_{max}}{\sigma_{min}})}$, and 2) VPSDE $\mathrm{d}\mathbf{x}_t = -\frac{1}{2}\beta(t)\mathbf{x}_t\,\mathrm{d}t + \sqrt{\beta(t)}\,\mathrm{d}\boldsymbol{\omega}_t$ with $\beta(t) = \beta_{min} + (\beta_{max} - \beta_{min})t$. With these diffusion schemes, VESDE has $\mu(t) \equiv 1$ with $\sigma^2(t) = \sigma_{min}^2\big[(\frac{\sigma_{max}}{\sigma_{min}})^{2t} - 1\big]$, and VPSDE has $\mu(t) = e^{-\frac{1}{2}\int_0^t \beta(s)\,\mathrm{d}s}$ with $\sigma^2(t) = 1 - e^{-\int_0^t \beta(s)\,\mathrm{d}s}$. We observe that the transformation of $\mathbf{y}_t = e^{\frac{1}{2}\int_0^t \beta(s)\,\mathrm{d}s}\mathbf{x}_t$ reduces VPSDE to VESDE of $\mathrm{d}\mathbf{y}_t = e^{\frac{1}{2}\int_0^t \beta(s)\,\mathrm{d}s}\sqrt{\beta(t)}\,\mathrm{d}\boldsymbol{\omega}_t$ by the Ito's lemma (Oksendal, 2013), so VPSDE and VESDE are indeed equivalent representations of the linear diffusion. Given their equivalence under the suggested transformation, we build our model based on VESDE as a default diffusion otherwise stated.

### 2.1 EFFECT OF DIFFUSION TIME ON DIFFUSION MODEL

**Density Estimation** In practice, previous diffusion models truncate the diffusion time from $[0, T]$ to $[\epsilon, T]$ to stabilize the training. Figure 1 emphasizes the loss contribution to the NELBO at $t \to 0$, which signifies lowering $\epsilon$ in the practice. Moreover, Figure 2 shows the cumulative loss of Figure 1 integrated over $[\sigma, \sigma_{max}]$ by changing the integrating variable from $t$ to $\sigma$ in Eq. 2, and the high precision (i.e., $\sigma \approx 0$ or $t \approx 0$) has significant portion on the NELBO.

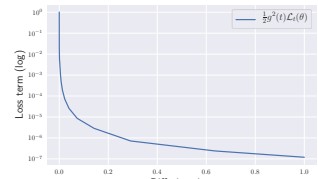

Figure 1: Loss contribution to NELBO by diffusion time.

$$\mathcal{L}(\boldsymbol{\theta}; g^2) = \frac{1}{2}\int_0^{\sigma_{max}} v\mathcal{L}_{\sigma^{-1}(v)}(\boldsymbol{\theta})\,\mathrm{d}v. \qquad (2)$$

Therefore, lowering $\sigma_{min} = \sigma(\epsilon)$ guarantees the tighter NELBO given its large portion at $t = \epsilon$. Empirically, NCSN++ (Song et al., 2020) performs 3.45 for NELBO trained with $\sigma_{min} = 10^{-2}$ on CIFAR-10, but this NELBO is tightened up to 3.04 by lowering $\sigma_{min}$ to $10^{-3}$ in our experiments.

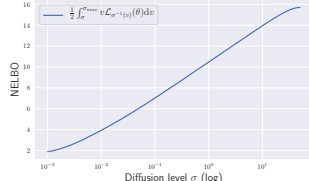

Figure 2: NELBO by diffusion level.

**Sample Generation** While we discussed the value of the loss at $t \to 0$, both fron and end diffusion time contribute to the sample quality in two distinctive ways.

*End Time* $(t \to T)$ : The diffusion model creates a fake sample by reverting the diffusion process with the score estimation. Figure 3 shows two contrastive image generations: the realistic image at the first row, and the unrealistic image at the second row with curly hair on its forehead. This unwanted hair is synthesized at the early steps of the reverse diffusion, which correspond to the end time of the forward diffusion. Therefore, Figure 3 implies that the overall sample quality is determined on the range of the end diffusion time.

Figure 3: Generative process.

*Front Time ($t \rightarrow 0$)* : The sample generation in Figure 4 shows that the local fidelity heavily depends upon the denoising process at the front of the diffusion time. This intuitive result is grounded by theory: observe that the random variable $\|\mathbf{x}_t - \mathbf{x}_0\|_2^2 / \sigma^2(t)$ follows the $\chi^2$-distribution of degree $d$, where $d$ is the data dimension. If the score network exactly esti-

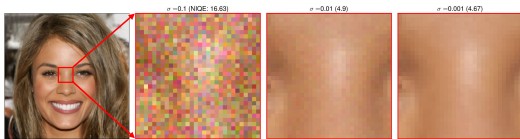

Figure 4: Generated images from CelebA-HQ $256 \times 256$. Image quality (NIQE (Mittal et al., 2012)) improves as the denoising proceeds.

mates the data score, the generated image at time $t$ would follow $\mathbf{x}_t$, and the average distance between the real sample ($\mathbf{x}_0$) with the fake sample ($\mathbf{x}_\epsilon$) becomes $\mathbb{E}_{p_r(\mathbf{x}_0)} \mathbb{E}_{p_{0\epsilon}(\mathbf{x}_\epsilon|\mathbf{x}_0)}[\|\mathbf{x}_\epsilon - \mathbf{x}_0\|_2^2] = \sigma_{min}^2 d$. Therefore, even if the score network perfectly estimates the data score, this leads that minimizing $\sigma_{min}$ acts in a crucial way for the pixel-wise precision.

From these observations, we identified that the front diffusion time is critical in generating the high resolution image for the pixel-wise details, and we also acknowledge that the end diffusion time contributes to the overall image structure. Now, our question becomes whether the existing models can learn the data score at both front and end diffusion time.

## 2.2 PROBLEMS OF TRAINING THE DIFFUSION MODEL WITH HIGH PRECISION

**Unbounded Score** Figure 5 presents that the data score is unbounded as the diffusion level, $\sigma^2(t)$, diminishes. This requires the score parametrization to handle this unboundedness. For instance, the parametrization of $\mathbf{s}_\theta(\mathbf{x}_t, \sigma(t))$ yields the failure of the score estimation at $t = 10^{-4}$ on the toy example in Figure 6.

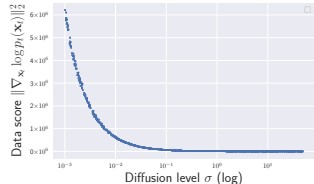

Figure 5: Unbounded score.

Theoretically, Lemma 1 states that the score estimation with $\mathbf{s}_\theta(\mathbf{x}_t, t)$ (DDPM) or $\mathbf{s}_\theta(\mathbf{x}_t, \sigma(t))$ (NCSN) fails if the data score is unbounded. See Appendix A for the proof.

**Lemma 1.** *Let $\mathcal{H}_{[0,T]} = \{\mathbf{s} : \mathbb{R}^d \times [0,T] \rightarrow \mathbb{R}^d, \mathbf{s} \text{ is locally Lipschitz}\}$. Suppose a continuous vector field $\mathbf{v}$ defined on a subset $U$ of a compact manifold $M$ (i.e., $\mathbf{v} : U \subset M \rightarrow \mathbb{R}^d$) is unbounded, then there exists no $\mathbf{s} \in \mathcal{H}_{[0,T]}$ such that $\lim_{t \rightarrow 0} \mathbf{s}(\mathbf{x}, t) = \mathbf{v}(\mathbf{x})$ a.e. on $U$.*

The score network, $\mathbf{s}_\theta$, is an element in $\mathcal{H}_{[0,T]}$ for *any* neural architecture as long as we put $t$ in the network, so Lemma 1 indicates that the vanilla score network *cannot* estimate an unbounded data score.

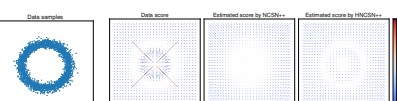

Figure 6: Score estimation of NCSN and HNCSN on a toy dataset.

**Training Instability** Training the diffusion time on the front range is advantageous on the creation of high-quality samples. However, training the diffusion model with lowered $\sigma_{min}$ induces the numerical instability, which eventually harms the overall sample quality.

To analyze the logic of the training instability, Figure 7 illustrates the Monte-Carlo samples of the diffusion loss, which blows up as $t \rightarrow 0$ (or $\sigma \rightarrow 0$). This leads that the diffusion loss is dominated by the diffusion time with the front range in practice, and the loss at the end diffusion time is barely counted in the gradient signal. This imbalanced loss brings the immatured score estimation at the end diffusion time, which ruins the overall sample shape.

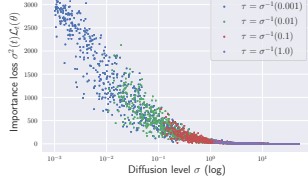

Figure 7: Imbalanced Monte-Carlo losses.

**Limitation of Diffusion Process** Continuous VESDE has its diffusion variance to be $\sigma^2(t) = \sigma_{min}^2 \left[ \left(\frac{\sigma_{max}}{\sigma_{min}}\right)^{2t} - 1 \right]$. As introduced in Song and Ermon (2020), VESDE has been suggested in its original discrete version based on the geometric property for its smooth transition of the distributional shift. Mathematically, the variance is geometric if $\frac{d}{dt} \log \sigma^2(t)$ is a constant, but the continuous VESDE breaks this geometric progression as in Figure 8.

To make it geometric, Song et al. (2020) approximates the VESDE diffusion process by the transition probability of $p_{0t}(\mathbf{x}_t|\mathbf{x}_0) =$

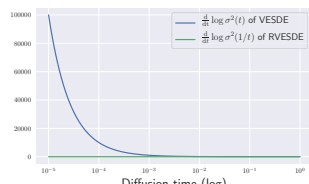

Figure 8: VESDE violates the geometric progression.

$\mathcal{N}\big(\mathbf{x}_t; \mathbf{x}_0, \sigma_{min}^2(\frac{\sigma_{max}}{\sigma_{min}})^{2t}\mathbf{I}\big)$, so this approximate diffusion process satisfies the geometric progression. However, this approximation leads that $\mathbf{x}_t$ is not converging to $\mathbf{x}_0$ in distribution because $\sigma_{min}^2(\frac{\sigma_{max}}{\sigma_{min}})^{2t} \to \sigma_{min}^2 > 0$ as $t \to 0$.

**Proposition 1.** *There is no SDE that has the stochastic process $\{\mathbf{x}_t\}_{t\in[0,T]}$, defined by a transition probability $p_{0t}(\mathbf{x}_t|\mathbf{x}_0) = \mathcal{N}(\mathbf{x}_t; \mathbf{x}_0, \sigma_{min}^2(\frac{\sigma_{max}}{\sigma_{min}})^{2t}\mathbf{I})$, as the solution.*

Proposition 1 indicates that if we approximate the diffusion process by $p_{0t}(\mathbf{x}_t|\mathbf{x}_0) = \mathcal{N}\big(\mathbf{x}_t; \mathbf{x}_0, \sigma_{min}^2(\frac{\sigma_{max}}{\sigma_{min}})^{2t}\mathbf{I}\big)$, then the reverse diffusion cannot be modeled by a reverse SDE. Therefore, VESDE loses its theoretic ground on the continuation of the discrete Markov chain.

## 3 HIGH PRECISION SCORE-BASED DIFFUSION MODELS

Given the three problems of unbounded score, training instability, and limited diffusion process, this section improves the diffusion model in three corresponding aspects. Section 3.1 introduces the theory-driven new parametrization, called HNCSN and HDDPM, that enables the successful score estimation for the diffusion model with small $\sigma_{min}$. Section 3.2 proposes the practical technique, ST-trick, which could estimate the score network at the end of the diffusion time without being dominated by the front diffusion. Section 3.3 suggests the new SDE, RVESDE, which develops the VESDE suitable for small $\sigma_{min}$.

### 3.1 PARAMETRIZATION MODEL FOR UNBOUNDED SCORE

As presented in Lemma 1, the score estimation of baseline models (NCSN/DDPM) fails because the locally Lipschitz with respect to the time argument does not hold anymore at $t = 0$. From this, we make the time argument extendable to the infinity at $t = 0$. To proceed, let us define $\eta : t \mapsto \mathbb{R}$ as a function that satisfies $\eta(t) \to \infty$ or $-\infty$ as $t \to 0$. Then, the parametrization of $\mathbf{s}_{\boldsymbol{\theta}}(\mathbf{x}_t, \eta(t))$ would successfully estimate the data score by Proposition 2.

**Proposition 2.** *Let $\mathcal{H}_{[1,\infty)} = \{\mathbf{s} : \mathbb{R}^d \times [1, \infty) \to \mathbb{R}^d, \ \mathbf{s} \ \text{is locally Lipschitz}\}$. Suppose a continuous vector field $\mathbf{v}$ defined on a $d$-dimensional open subset $U$ of a compact manifold $M$ is unbounded, and the projection of $\mathbf{v}$ on each axis is locally integrable. Then, there exists $\mathbf{s} \in \mathcal{H}_{[1,\infty)}$ such that $\lim_{\eta \to \infty} \mathbf{s}(\mathbf{x}, \eta) = \mathbf{v}(\mathbf{x})$ a.e. on $U$.*

Proposition 2 becomes the foundation of the unbounded parametrization, and the introduction of $\eta$ is the key to enable the estimation of the unbounded data score. Therefore, we introduce the new parametrization for the high precision score network as $\mathbf{s}_{\boldsymbol{\theta}}(\mathbf{x}_t, \eta(t))$, rather than $\mathbf{s}_{\boldsymbol{\theta}}(\mathbf{x}_t, t)$ (DDPM) or $\mathbf{s}_{\boldsymbol{\theta}}(\mathbf{x}_t, \sigma(t))$ (NCSN).

Table 1: Instantiation of the unbounded parametrization from Proposition 1. See Appendix B for the choice of $c_1, c_2, \sigma_0$.

| Model | $\eta(t)$ |
|---|---|
| HNCSN | $\eta(t) := \begin{cases} \log \sigma(t) & \text{if } \sigma(t) \geq \sigma_0 \\ -\frac{c_1}{\sigma(t)} + c_2 & \text{if } \sigma(t) < \sigma_0 \end{cases}$ |
| HDDPM | $\eta(t) := \int \frac{g^2(t)}{\sigma^2(t)} \, \mathrm{d}t$ |

Table 1 presents example parametrizations following Proposition 2. We suggest HNCSN with $\eta(t)$ as the mixture of $\log \sigma(t)$ at the end diffusion time and $\frac{1}{\sigma(t)}$ at the front diffusion time, in which the inverse function is beneficial on estimating the rapidly blowing data score by orders of $\frac{1}{\sigma}$. Note that Song and Ermon (2020) proposed the score parametrization of $\frac{\mathbf{s}_{\boldsymbol{\theta}}(\mathbf{x}_t)}{\sigma(t)}$, and Jolicoeur-Martineau et al. (2020) showed that the optimal score function becomes $\mathbf{s}^*(\mathbf{x}_t, t) = \frac{\mathbb{E}_{\mathbf{x}_0|\mathbf{x}_t}[\mathbf{x}_0] - \mathbf{x}_t}{\sigma^2(t)} \approx \frac{\mathbf{z}}{\sigma(t)}$, where $\mathbf{z} \sim \mathcal{N}(0, \mathbf{I})$. Therefore, we suggest HNCSN to put $\frac{1}{\sigma(t)}$ in place of $\log \sigma(t)$ at the front time.

Before we choose optimal $\eta$ for DDPM, observe that DDPM with the importance sampling has $\frac{g^2(t)}{\sigma^2(t)}$ as the importance weight, and this weight forces the sampled diffusion time highly concentrated on $t \approx \epsilon$. Since a small difference between two diffusion times near $t_1, t_2 \approx \epsilon$ results in a large deviation of two data scores in their scales, DDPM cannot approximate this extreme scale deviation with the vanilla parametrization of $\mathbf{s}_{\boldsymbol{\theta}}(\mathbf{x}_t, t)$ by Proposition 2.

We propose HDDPM with $\eta(t) = \int \frac{g^2(t)}{\sigma^2(t)} \, \mathrm{d}t$ as the antiderivative of the importance weight. For instance, HDDPM with VPSDE has its antiderivative as $\int \frac{g^2(t)}{\sigma^2(t)} \, \mathrm{d}t = \log\big(1 - e^{-\int_0^t \beta(s) \, \mathrm{d}s}\big) +$

$\int_0^t \beta(s)\,\mathrm{d}s$. This antiderivative diverges to $-\infty$ as $t \to 0$ for both VPSDE and subVPSDE, so it satisfies the condition of Proposition 2.

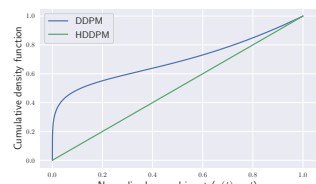

The choice of such $\eta(t)$ for HDDPM helps the enhanced score estimation near $t \approx \epsilon$ because the input of $\eta(t)$ is distributed uniformly when we draw samples from the importance weight. Concretely, when the sampling distribution on the diffusion time is given by $p(t) \propto \frac{g^2(t)}{\sigma^2(t)}$, the $\eta$-distribution from the importance sampling becomes $p(\eta) \propto 1$, which is depicted in Figure 9. Therefore, this uniform distribution on $\eta(t)$ in HDDPM guarantees the high precision to the score estimation.

Figure 9: Cumulative density function of $t$ and $\eta$.

## 3.2 Soft Truncation Trick for Practical Score Estimation

While the unbounded parametrization suggested in Section 3.1 provides the theoretic guarantee of the score estimation, this section proposes a practical optimization method for the high precision score estimation. In order to successfully train the score network without sacrificing the global fidelity, we detour the training instability by introducing the Soft Truncation (ST) trick that is taking a distinctive diffusion truncation for every mini-batch. In formula,

$$\mathcal{L}_{ST}(\boldsymbol{\theta}; \lambda, \mathbb{P}) = \mathbb{E}_{\mathbb{P}(\tau)}[\mathcal{L}(\boldsymbol{\theta}; \lambda, \tau)],$$

where $\mathbb{P}(\tau)$ is the prior distribution for the diffusion truncation. In every mini-batch update, we optimize $\mathcal{L}(\boldsymbol{\theta}; \lambda, \tau)$ with the sampled $\tau \sim \mathbb{P}(\tau)$ from the prior. Figure 7 illustrates four cases: blue dots with $\tau = \epsilon$, green dots with $\tau = \sigma^{-1}(0.01)$, etc. With ST-trick, a mini-batch with high truncation of $\tau = \sigma^{-1}(1)$ (purple) focuses on the score estimation at large $t$, and mini-batches with medium $\tau = \sigma^{-1}(0.1)$ (green) and $= \sigma^{-1}(0.01)$ (red) focus on the middle region of $[\epsilon, T]$. In consequence, ST-trick trains the diffusion model by partitioning its diffusion time for each mini-batch update.

The ST-trick with $\mathbb{P}(\tau) = \delta_\epsilon(\tau)$ unconditionally selects $\epsilon$ for every batch update, so it is equivalent with the original optimization of $\mathcal{L}(\boldsymbol{\theta}; \lambda, \epsilon)$. Although it is free to choose $\mathbb{P}$, we find that the reciprocal functions work best in practice. Observing that $\frac{1}{\tau^k} 1_{[\epsilon,T]}(\tau)/Z \to \delta_\epsilon(\tau)$ as $k \to \infty$ where $Z$ is the normalizing constant, the recipro-

Table 2: Example pairs of $(\mathbb{P}, \lambda_{\mathbb{P}})$.

|  | $\lambda$ | $k$ | $\mathbb{P}$ | $\lambda_{\mathbb{P}}$ |
|---|---|---|---|---|
| RVE | $\lambda(t)$ | $\infty$ | $\delta_\epsilon(\tau)$ | $\lambda(t)$ |
|  | $\lambda(t)$ | $2$ | $\frac{1}{\tau^2} 1_{[\epsilon,T]}(\tau)$ | $(1 - \frac{\epsilon}{t})\lambda(t)$ |
| VP | $\lambda(t)$ | $\infty$ | $\delta_\epsilon(\tau)$ | $\lambda(t)$ |
|  | $\lambda(t)$ | $1$ | $\frac{1}{\tau} 1_{[\epsilon,T]}(\tau)$ | $(1 - \log_\epsilon(t))\lambda(t)$ |

cal functions are connected to the original optimization. We enumerate few examples of $k = 2$ and $k = 1$ in Table 2 for RVESDE and VPSDE, respectively. Note that ST-trick is independent to the choice on the weighting function $\lambda$ for the batch-wise optimization loss, $\mathcal{L}(\boldsymbol{\theta}; \lambda, \tau)$.

**Theoretic Analysis of ST-trick** The ST-loss reduces the original diffusion loss to $\mathcal{L}_{ST}(\boldsymbol{\theta}; \lambda, \mathbb{P}) = \mathcal{L}(\boldsymbol{\theta}; \lambda_{\mathbb{P}}, \epsilon)$ with $\lambda_{\mathbb{P}}(t) := \left( \int_0^t \mathbb{P}(\tau)\,\mathrm{d}\tau \right) \lambda(t)$ in expectation by the Fubini theorem (Stein and Shakarchi, 2009). While the ST-loss in average is identical to the original diffusion loss, the major difference lies on that 1) $\mathcal{L}(\boldsymbol{\theta}; \lambda_{\mathbb{P}}, \epsilon)$ optimizes the diffusion loss in its averaged form and 2) $\mathcal{L}_{ST}(\boldsymbol{\theta}; \lambda, \mathbb{P})$ with ST-trick optimizes the diffusion loss with the Monte-Carlo sampled truncation, $\tau$.

To inspect how soft truncation affects to the optimization, let us assume $\lambda(t) = g^2(t)$. Then, we observe that the theoretic analysis previously suggested in Song et al. (2021) is directly extendable to any $\tau \in (0, T]$ by simply switching the positive truncation time, $\tau$, in place of the zero time:

$$\mathbb{E}_{p_\tau(\mathbf{x}_\tau)}\big[ -\log p_{\boldsymbol{\theta},\tau}(\mathbf{x}_\tau) \big] \leq \mathcal{L}(\boldsymbol{\theta}; g^2, \tau),$$

where $p_{\boldsymbol{\theta},\tau}$ is the marginal probability distribution at time $\tau$ of the reverse (generative) process that starts from the prior distribution and flows backwards in time, and $p_\tau$ is the marginal probability density at time $\tau$ of the forward diffusion process. Then, the above inequality is equivalent to

$$D_{KL}(p_\tau \| p_{\boldsymbol{\theta},\tau}) \leq \mathcal{L}(\boldsymbol{\theta}; g^2, \tau),$$

after omitting constants independent of $\boldsymbol{\theta}$ for the above two inequalities. This KL bound means that our ST-trick simply ignores the time range beneath $\tau$, and optimizes the proxy of the KL divergence between the forward and the reverse diffusion at time $\tau$. On the other hand, no such interpretation can be applied to the diffusion loss without ST-trick, $\mathcal{L}(\boldsymbol{\theta}; \lambda_{\mathbb{P}}, \epsilon)$.

### 3.3 RECIPROCAL VARIANCE EXPLODING SDE FOR SMALL TRUNCATION

This section introduces RVESDE whose variance, $\sigma^2(t)$, is geometric by its nature. RVESDE is defined as the form of $d\mathbf{x}_t = g(t)\,d\mathbf{w}_t$ with

$$
g(t) := \begin{cases} \sigma_{max}\big(\frac{\sigma_{min}}{\sigma_{max}}\big)^{\frac{\epsilon}{t}} \frac{\sqrt{2\epsilon \log\big(\frac{\sigma_{max}}{\sigma_{min}}\big)}}{t} & \text{if } t > 0, \\ 0 & \text{if } t = 0.^1 \end{cases}
$$

The transition probability of RVESDE becomes $p_{0t}(\mathbf{x}_t|\mathbf{x}_0) = \mathcal{N}\big(\mathbf{x}_t; \mathbf{x}_0, \sigma_{max}^2\big(\frac{\sigma_{min}}{\sigma_{max}}\big)^{\frac{2\epsilon}{t}}\mathbf{I}\big)$. Recall that VESDE assumes $g(t) = \sigma_{min}\big(\frac{\sigma_{max}}{\sigma_{min}}\big)^t \sqrt{2\log\big(\frac{\sigma_{max}}{\sigma_{min}}\big)}$ and $p_{0t}(\mathbf{x}_t|\mathbf{x}_0) = \mathcal{N}\big(\mathbf{x}_t; \mathbf{x}_0, \sigma_{min}^2\big[\big(\frac{\sigma_{max}}{\sigma_{min}}\big)^{2t} - 1\big]\mathbf{I}\big)$. As presented in Figure 8, RVESDE attains the geometric property at the expense of having reciprocated time, $1/t$. We could successfully model the forward diffusion process with RVESDE without any approximation, and this exact modeling of the forward process enables applying the stochastic calculus to compute the log-likelihood or to generate new samples.

It is common to apply the importance sampling (Song et al., 2021) in order to minimize the Monte-Carlo variance on the diffusion loss as illustrated in Figure 13 (c). RVESDE is advantageous on VE/VPSDEs that the importance sampling is easy to implement: the diffusion loss with RVESDE becomes

$$
\mathcal{L}(\boldsymbol{\theta}; g^2, \epsilon) = \frac{1}{2}\int_\epsilon^T g^2(t)\mathcal{L}_t(\boldsymbol{\theta})\,dt = \epsilon \log\Big(\frac{\sigma_{max}}{\sigma_{min}}\Big)\int_{\frac{1}{T}}^{\frac{1}{\epsilon}} \sigma^2(s^{-1})\mathcal{L}_{s^{-1}}(\boldsymbol{\theta})\,ds,
$$

by substituting the integrating variable into the reciprocated time. Therefore, the importance sampling for RVESDE is equivalent to drawing the diffusion time uniformly on a reciprocated interval, $[\frac{1}{T}, \frac{1}{\epsilon}]$.

## 4 EXPERIMENTS

This section empirically studies our suggestions on benchmark datasets, including CIFAR-10 (Krizhevsky et al., 2009), STL-10 (Coates et al., 2011)[2] CelebA (Liu et al., 2015) $64 \times 64$, CelebA-HQ (Karras et al., 2017) $256 \times 256$, LSUN (Yu et al., 2015) Bedroom/Church $256 \times 256$, and FFHQ (Karras et al., 2019). We compute *bits-per-dim*, proportional to NLL, by applying the Instantaneous Change of Variable Chen et al. (2018); Song et al. (2020) to the probability flow (Song et al., 2020) with the uniform dequantization (Theis et al., 2015; Hoogeboom et al., 2020), rather than the variational dequantization (Ho et al., 2019) in order to sidestep the need of training an auxiliary flow network. Also, we use the clean-FID introduced in Parmar et al. (2021) for computing the FID (Heusel et al., 2017) score with the predictor-corrector sampler (Song et al., 2020). Training and evaluation details are available in Appendix B.

### 4.1 QUANTITATIVE RESULTS

Table 3 compares HNCSN++ (RVE) with ST-trick against the current best generative models, and Table 3 shows that our model establishes the state-of-the-art performances. On CIFAR-10, we observe that our model improves NCSN++ (VE) in NLL and IS at the expense of sacrificing FID slightly. In particular, our model surpasses the previous best IS performance (StyleGAN2-ADA+Tuning (Karras et al., 2020)) by performing 10.11 on CIFAR-10. HNCSN++ trained on CelebA is the best performer in terms of the FID performance, but CR-NVAE (Sinha and Dieng, 2021) outperforms our model in NLL. However, we emphasize that HDDPM++ in Table 6 with ST-trick (1.52) outperforms CR-NVAE (1.86) in NLL by far. In particular, HNCSN++ with high-dimensional CelebA-HQ $256 \times 256$ performs the state-of-the-art in FID out of the baselines in the diffusion models (NCSN++) and the GAN models (PGGAN (Karras et al., 2017)). Finally, HNCSN++ on STL-10 dataset largely outperforms the baselines by improving the state-of-the-art FID from 15.17 (Park and Kim, 2021) to 7.71, and IS from 11.01 (Park and Kim, 2021) to 13.43.

In Table 3, when computing the Inception Score (IS) (Salimans et al., 2016), there is a minor discrepancy between likelihood-based models and likelihood-free models. IS-50k in diffusion models

---

[1] The existence and uniqueness of solution for RVESDE is guaranteed by Theorem 5.2.1 in Oksendal (2013).

[2] We downsize the dataset from $96 \times 96$ to $48 \times 48$ following Jiang et al. (2021); Park and Kim (2021).

Table 3: Performance comparisons on benchmark datasets. The boldfaced numbers present the best performance, and the underlined numbers present the second-best performance.

| Model | CIFAR10 32 × 32 | | | CelebA 64 × 64 | | CelebA-HQ 256 × 256 (8-bits) | STL-10 48 × 48 | |
|---|---|---|---|---|---|---|---|---|
| | NLL ($\downarrow$) | FID ($\downarrow$) | IS ($\uparrow$) | NLL | FID | FID | FID | IS |
| HNCSN++ (RVE) + ST | 3.04 | 2.33 | **10.11** | 1.93 | **1.92** | **7.16** | **7.71** | **13.43** |
| **Likelihood-based Models** | | | | | | | | |
| CR-NVAE (Sinha and Dieng, 2021) | **2.51** | - | - | **1.86** | - | - | - | - |
| LSGM (FID) (Vahdat et al., 2021) | 3.43 | **2.10** | - | - | - | - | - | - |
| DenseFlow-74-10 (Grcić et al., 2021) | 2.98 | - | - | 1.99 | - | - | - | - |
| Gamma Distribution DDIM (Nachmani et al., 2021) | - | - | - | - | 2.92 | - | - | - |
| VDM (Kingma et al., 2021) | 2.65 | - | - | - | - | - | - | - |
| NCSN++ cont. (deep, VE) (Song et al., 2020) | 3.45 | 2.20 | 9.89 | 2.39 | 3.95 | 7.23 | - | - |
| DDPM++ cont. (deep, sub-VP) (Song et al., 2020) | 2.99 | 2.41 | 9.57 | - | - | - | - | - |
| ScoreFlow (cont. norm. flow) (Song et al., 2021) | 2.74 | 5.70 | - | - | - | - | - | - |
| Improved DDPM ($L_{simple}$) (Nichol and Dhariwal, 2021) | 3.37 | 2.90 | - | - | - | - | - | - |
| **Likelihood-free Models** | | | | | | | | |
| StyleGAN2-ADA+Tuning (Karras et al., 2020) | - | 2.92 | 10.02 | - | - | - | - | - |
| Styleformer (Park and Kim, 2021) | - | 2.82 | 9.94 | - | 3.66 | - | 15.17 | 11.01 |
| PGGAN (Karras et al., 2017) | - | - | 8.8 | - | - | 8.03 | - | - |
| TransGAN (Jiang et al., 2021) | - | 9.26 | 9.02 | - | 5.01 | - | 18.28 | 10.43 |

(Song et al., 2020; Ho et al., 2020) computes the score with 50k generated images once, while IS-5k in GAN models (Karras et al., 2020; Park and Kim, 2021) computes the score 10 trials with independently generated 5k samples and report the performance as the average of scores. We report IS-50k following Song et al. (2020), but we note that IS-5k performs almost identical to IS-50k by performing 10.07 on CIFAR-10 and 13.34 on STL-10 in terms of IS-5k.

One particular observation in Table 3 is the IS performance (13.34) of STL-10 that exceeds the number of classes (10) of the dataset. We follow Park and Kim (2021), the current state-of-the-art model on STL-10, to train STL-10 with 105k images by aggregating the labeled (5k) and the unlabeled (100k) images (Jiang et al., 2021). Although the labeled images have 10 classes, the unlabeled images are sampled from a broader distribution of images, so the unlabeled dataset contains the other types of animals such as bears or rabbits, whose classes are different from the labeled classes. Therefore, a well-trained model with 105k images would perform IS that exceeds the number of labeled classes.

## 4.2 ABLATION STUDY

Throughout the ablation study, we train the model on CelebA as default otherwise stated.

**High Precision Parametrization** Table 4 presents the model performances after 0.5M of training iterations. First, NCSN++ with lowered $\sigma_{min}$ effectively reduces the performances on both NLL and FID, as anticipated in Section 2. Second, the new parametrization of HNCSN gives a slight performance gain compared to the original parametrization of NCSN. This performance gain comes from the better estimation for the data score, evidenced by Figure 13 (a). Third, the new parametrization of HDDPM largely improves the sample quality at the expense of slight NLL degradation.

Table 4: Ablation study for the new parametrization.

| Model | $\sigma_{min}$ | CelebA | |
|---|---|---|---|
| | | NLL | FID |
| NCSN++ (VE) | $10^{-2}$ | 2.39 | 3.95 |
| NCSN++ (VE) | $10^{-3}$ | 1.96 | 3.59 |
| HNCSN++ (VE) | $10^{-3}$ | 1.97 | 3.54 |
| HNCSN++ (RVE) | $10^{-3}$ | 1.97 | 3.36 |
| DDPM++ (VP, IS) | $10^{-3}$ | 1.53 | 5.31 |
| HDDPM++ (VP, IS) | $10^{-3}$ | 1.64 | 4.65 |

**Soft Truncation** Figure 10 compares the diffusion model with/without ST-trick in terms of the sample fidelity by training iteration. To compare in a fair setting, we train the model in a pair of comparative settings: first, we train the model with vanilla diffusion loss of $\mathcal{L}(\boldsymbol{\theta}; \lambda_{\mathbb{P}}, \epsilon)$ with $\lambda_{\mathbb{P}}(t) = (1 - \frac{\epsilon}{t})g^2(t)$; and second, we train according to the same loss with ST-trick of $\lambda(t) = g^2(t)$ and $\mathbb{P}(\tau) \propto \frac{1}{\tau^2}$. At the initial stage of training, the sample fidelity with ST-trick is worse than that without ST-trick because the model with ST-trick learns the data score of small diffusion slower. However, ST-trick effectively updates the score network as training proceeds,

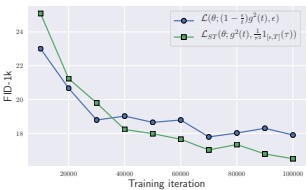

Figure 10: ST-trick improves the sample generation.

and Figure 10 shows that the model with ST-trick surpasses the model without ST-trick after the training.

Table 5 presents the performances after 0.5M of training steps. It compares the diffusion models with an identical weighting function, $(1 - \frac{\epsilon}{t})g^2$, with/without ST-trick, and it shows that ST-trick improves NLL from 1.98 to 1.93 and FID from 3.18 to 1.92, where 1.92 beats the current state-of-the-art FID (2.92) by far.

Table 5: Ablation study for ST-trick trained on HNCSN++ (RVE).

| $\mathbb{P}(\tau)$ | Optimization Loss | ST | CelebA NLL | FID |
|---|---|---|---|---|
| $\delta_\epsilon(\tau)$ | $\mathcal{L}(\boldsymbol{\theta}; g^2, \epsilon)$ | ✗ | 1.97 | 3.36 |
| $\delta_\epsilon(\tau)$ | $\mathcal{L}(\boldsymbol{\theta}; (1 - \frac{\epsilon}{t})g^2, \epsilon)$ | ✗ | 1.98 | 3.18 |
| $\frac{1}{\tau^2}1_{[\epsilon,T]}(\tau)$ | $\mathcal{L}_{ST}(\boldsymbol{\theta}; g^2, \mathbb{P})$ | ✓ | 1.93 | 1.92 |

Table 6 compares our ST-trick with the vanilla optimization of the hard truncation on weighting functions of $\lambda(t) = g^2(t)$ and $\lambda(t) = \sigma^2(t)$. Table 6 shows that ST-trick is effective on both sample generation and density estimation. In particular, for any model settings, we conclude that ST-trick is a highly effective method that significantly enhances the sample quality without any of model modification.

**RVESDE** Table 4 shows the efficacy of RVESDE on HNCSN++. Applying RVESDE improves the FID score while preserving the NLL performance, compared to VESDE on HNCSN++. To analyze the performance, recall that Proposition 2 implies that the inequality of NELBO in 1 does not hold anymore, so the NLL performance of VESDE loses its theoretic ground on bounding the log-likelihood. However, our RVESDE allows the inequality of NELBO, so the NLL performance in Table 4 is theoretically grounded. In addition, the experiments in Table 6 shows that RVESDE harmonizes with ST-trick better than VESDE.

Table 6: Ablation study for ST-trick.

| Model | $\lambda$ | ST | CelebA NLL | FID |
|---|---|---|---|---|
| NCSN++ (VE) | $\sigma^2$ | ✗ | 2.39 | 3.95 |
| | | ✓ | 2.41 | 2.68 |
| HNCSN++ (RVE) | $g^2$ | ✗ | 1.97 | 3.36 |
| | | ✓ | 1.93 | 1.92 |
| DDPM++ (VP) | $\sigma^2$ | ✗ | 1.79 | 3.03 |
| | | ✓ | 1.75 | 2.88 |
| HDDPM++ (VP) | $\sigma^2$ | ✗ | 1.78 | 3.23 |
| | | ✓ | 1.73 | **2.22** |
| DDPM++ (VP, IS) | $g^2$ | ✗ | 1.53 | 5.31 |
| | | ✓ | **1.52** | 4.50 |
| HDDPM++ (VP, IS) | $g^2$ | ✗ | 1.64 | 4.65 |
| | | ✓ | **1.52** | 4.45 |

**Optimal Precision** Table 7 finds that lowering $\sigma_{min}$ too small might harm the sample quality due to the training instability. We find that $\sigma_{min} = 10^{-3}$ is a sweet spot for the optimal sample quality. This tendency maintains even when we apply the training stabilization method, such as ST-trick.

Table 7: Ablation study for $\sigma_{min}$ trained on HNCSN++ (RVE).

| ST | $\sigma_{min}$ | CIFAR-10 NLL | FID-10k |
|---|---|---|---|
| ✗ | $10^{-3}$ | 2.96 | 7.04 |
| | $10^{-4}$ | 2.97 | 8.17 |
| | $10^{-5}$ | 2.97 | 8.29 |
| ✓ | $10^{-3}$ | 2.99 | 5.09 |
| | $10^{-4}$ | 2.97 | 5.54 |
| | $10^{-5}$ | 2.96 | 5.98 |

The reasoning of such opposite trend could be partially explained by how the data is saved. When we save the data, it is 8-bit quantized in order to downsize the file size, so it is enough to downscale $\sigma_{min}$ less than $1/256$ to fit the model distribution to the data distribution, which is approximately 0.004. Therefore, lowering $\sigma_{min}$ too small incurs severe numerical instability without gaining additional performance enhancement.

Empirical result in Figure 11 indicates that $\sigma_{min}$ of $10^{-3}$ is indeed enough to create high precision images on high-dimensional FFHQ $256 \times 256$ dataset. Figure 11 illustrates the histogram of the image-by-image quality metric, evaluated by Natural Image Quality Evaluator (NIQE) (Mittal et al., 2012), with 5k samples. Figure 11 implies that the denoised images up to the diffusion levels of 0.1 or 0.01 are not satisfactory in terms of this image-by-image metric. In fact, this discrepancy of the image-to-image quality metric explains the gain in FID when we lower $\sigma_{min}$ in Table 4, where FID is a distributional metric for sample quality.

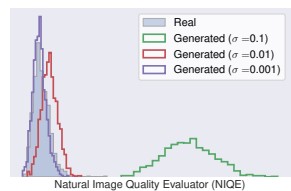

Figure 11: Optimal precision on FFHQ $256 \times 256$.

**High Dimensions** To clarify the modeling effect on high dimensional datasets, we train HNCSN++ (RVE) with ST-trick for $\sigma_{min} = 10^{-3}$ on datasets of $256 \times 256$ resolutions. After the training of 3.3M iterations, we draw 5k samples by stopping the reverse process at $\sigma = 10^{-1}/10^{-2}/10^{-3}$. When we denoise up to $\sigma = 10^{-2}$, Table 8 shows that our model largely improves the sample quality: HNCSN++ outperforms by reporting 13.52 and 14.20 in FID-5k compared to the NCSN++ of 16.42 and 29.14 on FFHQ $256 \times 256$ and LSUN Church $256 \times 256$, respectively. Such a huge discrepancy on high-dimensional datasets is attributed to the poorly synthesized images from NCSN++, which is evidenced by the right

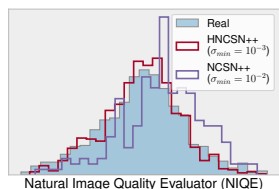

Figure 12: Image-by-image quality histogram on CelebA-HQ $256 \times 256$.

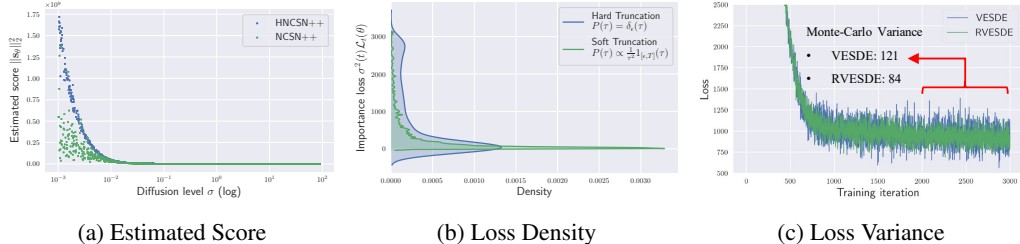

| (a) Estimated Score | (b) Loss Density | (c) Loss Variance |

Figure 13: Qualitative experimental results for (a) score estimation, (b) Monte-Carlo loss density of ST-trick, (c) Monte-Carlo loss variance without ST-trick.

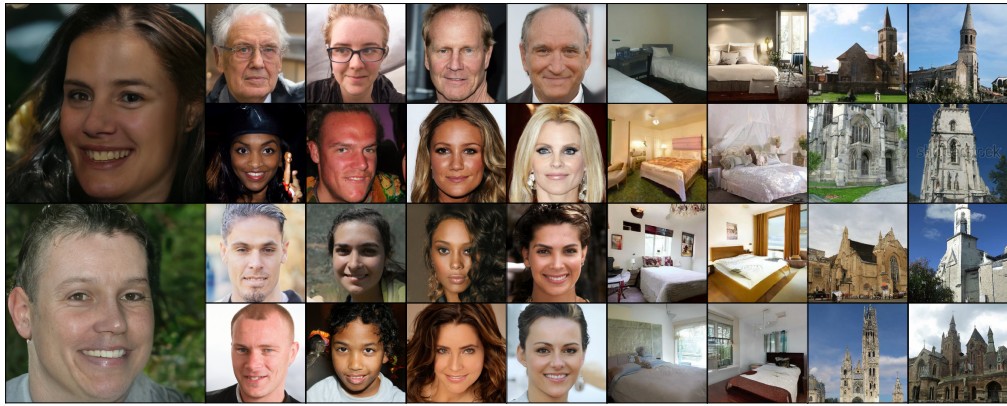

Figure 14: Generated images from FFHQ (1st column) / FFHQ 256 (2,3th columns) / CelebAHQ (4,5th columns) / LSUN Bedroom (6,7th columns) / LSUN Church (8,9th columns)

heavy tail in Figure 12. Also, as we denoise further from $10^{-2}$ to $10^{-3}$, the sample quality improves from 13.52 to 10.35 on FFHQ $256 \times 256$ and 14.20 to 13.75 on LSUN Church $256 \times 256$.

### 4.3 QUALITATIVE RESULTS

This section presents the qualitative experimental results for the proposed methods. Figure 13 (a) compares the score estimation for HNCSN++ and NCSN++ on CelebA. HNCSN++ succeeds in estimating the unbounded score, whereas NCSN++ shows poor estimation at the front diffusion time (or small diffusion level). Figure 13 (b) compares the diffusion model with/without ST-trick in terms of the Monte-Carlo loss density. As expected, the density is bimodal for the model without ST-trick, but it turns out that this heavy tail is largely improved by ST-trick. Figure 13 (c) illustrates that the Monte-Carlo loss variance is reduced with RVESDE, compared to VESDE. We attribute this variance reduction to the exact modeling of the diffusion process.

Table 8: Ablation study for denoising effect on high-dimensional datasets.

| Model | $\sigma$ | FFHQ FID-5k | Church FID-5k |
|---|---|---|---|
| HNCSN++ (RVE) + ST | $10^{-1}$ | 60.56 | 48.06 |
| | $10^{-2}$ | 13.52 | 14.20 |
| | $10^{-3}$ | 10.35 | 13.75 |
| NCSN++ | $10^{-2}$ | 16.42 | 29.14 |

Figure 14 shows the generated images from various benchmark datasets. We find that our model succeeds generating realistic images on high-dimensional datasets with resolution greater or equal to $256 \times 256$. See Appendix C for the full image generation results.

### 5 CONCLUSION

This paper tackles the problem of exploding data score with three points: 1) we provide a theory-driven parametrization for the high precision estimation; 2) we introduce a practical optimization method of the diffusion model parameters; and 3) we suggest a new type of SDE that enables to apply the stochastic calculus on small diffusion level. Applying the three improvements to NCSN/DDPM, we presented the high precision version of NCSN and DDPM, which excels in generating the high resolution images.

## 6 ETHICS STATEMENT

Potential risk from this work is the negative usage of the deep generative models, such as deep-fake images. Our contribution enables the creation of the high resolution virtual images, and we acknowledge that there is any chance of misuse for malicious purposes.

## 7 REPRODUCIBILITY STATEMENT

Complete proofs of the theoretic analyses in the main paper are provided in Appendix A. The code for the experiments would be released to the reviewers at the discussion phase through an anonymous repository for the reproducibility. Most of our code is built based on the released code of the previous work (Song et al., 2020) including the network structure and hyperparameters, as explained in Appendix B.

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

## A PROOFS OF LEMMA 1, PROPOSITION 1, AND PROPOSITION 2

**Lemma 1.** *Let $\mathcal{H}_{[0,T]} = \{\mathbf{s} : \mathbb{R}^d \times [0,T] \to \mathbb{R}^d, \mathbf{s}$ is locally Lipschitz$\}$. Suppose a continuous vector field $\mathbf{v}$ defined on a subset $U$ of a compact manifold $M$ (i.e., $\mathbf{v} : U \subset M \to \mathbb{R}^d$) is unbounded, then there exists no $\mathbf{s} \in \mathcal{H}_{[0,T]}$ such that $\lim_{t \to 0} \mathbf{s}(\mathbf{x},t) = \mathbf{v}(\mathbf{x})$ a.e. on $U$.*

*Proof.* Since $U$ is an open subset of a compact manifold $M$, $\|\mathbf{x}_1 - \mathbf{x}_2\| \leq \mathrm{diam}(M)$ for all $\mathbf{x}_1, \mathbf{x}_2 \in U$. Also, if $t_1, t_2 \in [0,T]$, $|t_1 - t_2|$ is bounded. Hence, the local Lipschitzness of $\mathbf{s}$ implies that there exists a positive $K > 0$ such that $\|s(\mathbf{x}_1, t_1) - s(\mathbf{x}_2, t_2)\| \leq K(\|\mathbf{x}_1 - \mathbf{x}_2\| + |t_1 - t_2|)$ for any $\mathbf{x}_1, \mathbf{x}_2 \in U$ and $t_1, t_2 \in [0,T]$. Therefore, for any $\mathbf{s} \in \mathcal{H}_{[0,T]}$, there exists $C > 0$ such that $\|\mathbf{s}(\mathbf{x},t)\| < C$ for all $\mathbf{x} \in U$ and $t \in [0,T]$, which leads no $\mathbf{s}$ that satisfies $\mathbf{s}(\mathbf{x},t) \to v(\mathbf{x})$ a.e. on $U$ as $t \to 0$. $\square$

**Proposition 2.** *Let $\mathcal{H}_{[1,\infty)} = \{\mathbf{s} : \mathbb{R}^d \times [1,\infty) \to \mathbb{R}^d, \mathbf{s}$ is locally Lipschitz$\}$. Suppose a continuous vector field $\mathbf{v}$ defined on a $d$-dimensional open subset $U$ of a compact manifold $M$ is unbounded, and the projection of $\mathbf{v}$ on each axis is locally integrable. Then, there exists $\mathbf{s} \in \mathcal{H}_{[1,\infty)}$ such that $\lim_{\eta \to \infty} \mathbf{s}(\mathbf{x}, \eta) = \mathbf{v}(\mathbf{x})$ a.e. on $U$.*

*Proof.* Let $h$ be a standard mollifier function. If $h_t(x) = t^{-n} h(\mathbf{x}/t)$, then $v_t := h_t * v$ converges to $v$ a.e. on $U$ as $t \to 0$ (Theorem 7-(ii) of Appendix C in Evans (1998)). Therefore, if we define $s(\mathbf{x}, \eta) := v_{1/\eta}(\mathbf{x})$ on the domain of $v_{1/\eta}(\mathbf{x})$ and $s(\mathbf{x}, \eta) := 0$ elsewhere, then $s(\mathbf{x}, \eta) = v_{1/\eta}(\mathbf{x}) \to v(\mathbf{x})$ a.e. on $U$ as $\eta \to \infty$.

Now, to show that $\mathbf{s}(\mathbf{x}, \eta)$ is locally Lipschitz, let $\tilde{M} \times [\underline{\eta}, \overline{\eta}]$ be a compact subset of $\mathbb{R}^n \times [1, \infty)$. From $\|\mathbf{s}(\mathbf{x}_1, \eta_1) - \mathbf{s}(\mathbf{x}_2, \eta_2)\| = \|v_{1/\eta_1}(\mathbf{x}_1) - v_{1/\eta_2}(\mathbf{x}_2)\| \leq \|v_{1/\eta_1}(\mathbf{x}_1) - v_{1/\eta_1}(\mathbf{x}_2)\| + \|v_{1/\eta_1}(\mathbf{x}_2) - v_{1/\eta_2}(\mathbf{x}_2)\|$, if there exists $K_1, K_2 > 0$ such that $\|v_{1/\eta_1}(\mathbf{x}_1) - v_{1/\eta_1}(\mathbf{x}_2)\| \leq K_1 \|\mathbf{x}_1 - \mathbf{x}_2\|$ and $\|v_{1/\eta_1}(\mathbf{x}_1) - v_{1/\eta_2}(\mathbf{x}_1)\| \leq K_2 |\eta_1 - \eta_2|$ for all $\mathbf{x}_1, \mathbf{x}_2 \in \tilde{M}$ and $\eta_1, \eta_2 \in [\underline{\eta}, \overline{\eta}]$, then $\mathbf{s}(\mathbf{x}, \eta) = v_{1/\eta}(\mathbf{x})$ is Lipschitz on $\tilde{M} \times [\underline{\eta}, \overline{\eta}]$.

First, since $v_{1/\eta}$ is infinitely differentiable on its domain (Theorem 7-(i) of Appendix C in Evans (1998)) and $\eta \in [\underline{\eta}, \overline{\eta}]$, there exists $K_1 > 0$ such that $\|v_{1/\eta}(\mathbf{x}_1) - v_{1/\eta}(\mathbf{x}_2)\| \leq K_1 \|\mathbf{x}_1 - \mathbf{x}_2\|$. Second, the mollifier satisfies the uniform convergence on any compact subset of $U$ (Theorem 7-(iii) of Appendix C in Evans (1998)), which leads that $\|v_{1/\eta_1}(\mathbf{x}) - v_{1/\eta_2}(\mathbf{x})\| \leq K_2 |\frac{1}{\eta_1} - \frac{1}{\eta_2}| = K_2 \frac{|\eta_1 - \eta_2|}{\eta_1 \eta_2} \leq K_3 |\eta_1 - \eta_2|$ for some $K_2, K_3 > 0$. Therefore, $\mathbf{s}$ becomes an element of $\mathcal{H}_{[1,\infty)}$. $\square$

## B IMPLEMENTATION DETAILS

Since the major contribution of this paper is developing the diffusion model in perspectives of theory and optimization method, we use the same model architecture introduced in Song et al. (2020).

### B.1 EXPERIMENTAL DETAILS

**Training** Throughout the experiments, we train our model with a learning rate of 0.0002, warmup of 5000 iterations, gradient clipping by 1. On the experiments for HNCSN++ with $\sigma_{min} = 10^{-3}$, we train the diffusion model by truncating $\sigma_{min}$ to $10^{-2}$ at the initial stage of training, and after the saturation in terms of the FID score, we begin to train the score network with the whole range of $[\sigma_{min}, \sigma_{max}]$. We find this way of training works best in practice for the training stability. We follow the convention of Song et al. (2020) to choose $\sigma_{max}$. Also, for DDPM-style model training, we use $\beta_{min} = 0.1$ and $\beta_{max} = 20$. For all experiments, we set $\epsilon = 10^{-5}$ and $T = 1$.

We stabilize the training by the Exponential Moving Average (EMA) (Song et al., 2020) with the rate of 0.999 on NCSN++/HNCSN++ and 0.9999 on DDPM++/HDDPM++. In addition, we stabilize the training by applying the dropout (Srivastava et al., 2014) with 0.1 rate. For the optimizer, we use the Adam optimizer (Kingma and Ba, 2014) with $(\beta_1, \beta_2) = (0.9, 0.999)$.

On datasets of resolution $32 \times 32$, we use the batch size of 128, which consumes about 48Gb GPU memory. On STL-10 with resolution $48 \times 48$, we use the batch size of 192, and on datasets of resolution $64 \times 64$, we experiment with 128 batch size. The batch size for the datasets of resolution

$256 \times 256$ is 40, which takes nearly 120Gb of GPU memory. On the dataset of $1024 \times 1024$ resolution, we use the batch size of 16, which takes around 120Gb of GPU memory. We use five NVIDIA RTX-3090 GPU machines to train the model exceeding 48Gb, and we use a pair of NVIDIA RTX-3090 GPU machines to train the model that consumes less than 48Gb.

**Evaluation** We apply the EMA with rate of 0.999 on NCSN++/HNCSN++ and 0.9999 on DDPM++/HDDPM++. For the density estimation, we obtain the NLL performance by the Instantaneous Change of Variable (Song et al., 2020; Chen et al., 2018). For the sampling, we apply the Predictor-Corrector (PC) algorithm introduced in Song et al. (2020). We set the signal-to-noise ratio as 0.16 on $32 \times 32$ datasets, 0.17 on $48 \times 48$ and $64 \times 64$ datasets, 0.075 on $256 \times 256$ sized datasets, and 0.15 on $1024 \times 1024$. On datasets less than $256 \times 256$ resolution, we iterate 1,000 steps for the PC sampler, while we apply 2,000 steps on the other high-dimensional datasets. Throughout the experiments, we use the reverse diffusion (Song et al., 2020) for the predictor algorithm and the annealed Langevin dynamics (Welling and Teh, 2011) for the corrector algorithm.

We compute the FID score (Song et al., 2020) based on the modified Inception V1 network[3] using the tensorflow-gan package for CIFAR-10 dataset, and we use the clean-FID (Parmar et al., 2021) based on the Inception V3 network (Szegedy et al., 2016) for the remaining datasets. We note that FID computed by Parmar et al. (2021) reports a higher FID score compared to the original FID calculation[4].

### B.2 CHOICE OF $\eta(t)$

**HNCSN** As described in Section 3.1, HNCSN assumes the score parametrization of $\mathbf{s}_{\boldsymbol{\theta}}(\mathbf{x}_t, \eta(t))$ with $\eta(t)$ being the mixture of $\log \sigma(t)$ and $\frac{1}{\sigma(t)}$. Despite of the clear advantage of $\log \sigma(t)$ that the distribution of $p(\sigma)$ is uniform in VE/RVESDEs, the drastic scale variation at $t \approx 0$ prevents the enhanced score estimation. We empirically find that the reciprocal function of $\frac{1}{\sigma(t)}$ works best on the estimation at the front diffusion time, and we combine these two functions to create a continuously differentiable $\eta(t)$ by

$$\eta(t) := \begin{cases} \log \sigma(t) & \text{if } \sigma(t) \geq \sigma_0 \\ -\frac{c_1}{\sigma(t)} + c_2 & \text{if } \sigma(t) < \sigma_0, \end{cases}$$

where $c_1, c_2$ and $\sigma_0$ are the hyperparameters. By acknowledging the parametrization of $\log \sigma(t)$ introduced in Song et al. (2020), we choose $\sigma_0$ as 0.01. Also, to satisfy the continuously differentiability of $\eta(t)$, the choice of two hyperparameters $c_1$ and $c_2$ satisfies a system of equations with degree 2, so $c_1$ and $c_2$ are fully determined with this system of equations.

**HDDPM** Looking at the antiderivative $\int \frac{g^2(t)}{\sigma^2(t)} \, dt = \log\left(1 - e^{-\int_0^t \beta(s) \, ds}\right) + \int_0^t \beta(s) \, ds = \log \sigma^2(t) + \int_0^t \beta(s) \, ds$, the unbounded function reduces to $\eta(t) = 2 \log \sigma(t) + \int_0^t \beta(s) \, ds$. Therefore, HDDPM resembles the temporal parametrization of $\log \sigma(t)$ (Song et al., 2020) at $t \approx 0$ because the second term, $\int_0^t \beta(s) \, ds$, is nearly zero at $t \approx 0$.

## C ADDITIONAL EXPERIMENTAL RESULTS

Figure 15 shows how images are created from the trained model on 256-dimensional datasets, and Figures from 16 to 21 present randomly generated samples of the trained model. Also, Tables 9 and 10 present the additional experimental results in terms of FID on LSUN Bedroom and FFHQ $256 \times 256$.

---

[3]`https://tfhub.dev/tensorflow/tfgan/eval/inception/1`
[4]See `https://github.com/GaParmar/clean-fid` for the detailed experimental results.

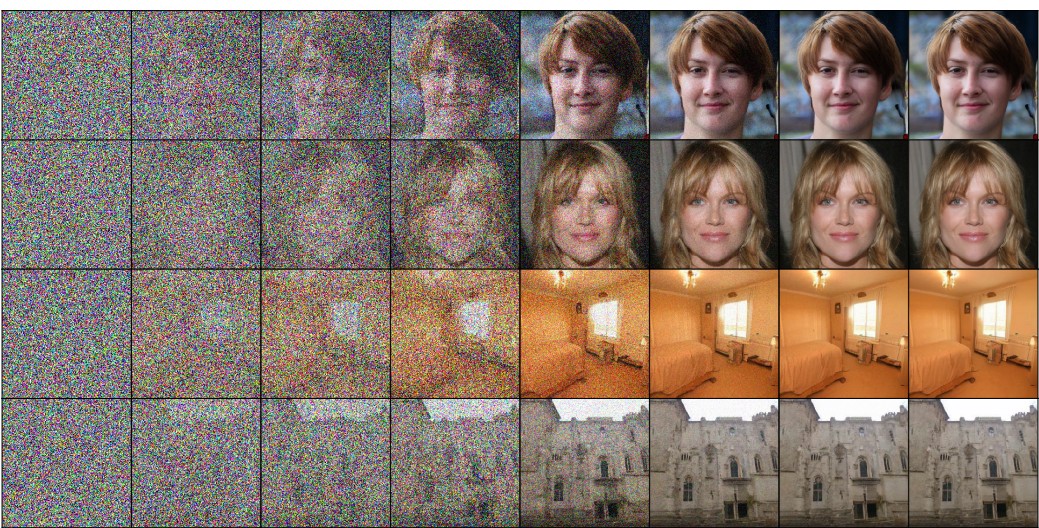

Figure 15: Image generation by denoising via predictor-corrector sampler.

Table 9: LSUN Bedroom

| Model | Bedroom 256 × 256 FID |
|---|---|
| UDM (RVE) + ST | 4.57 |
| **Likelihood-based Models** | |
| ADM Dhariwal and Nichol (2021) | **1.90** |
| DDPM Ho et al. (2020) | 4.90 |
| **Likelihood-free Models** | |
| StyleGAN Karras et al. (2019) | 2.65 |
| INR-GAN-bil Skorokhodov et al. (2020) | 4.95 |
| COCO-GAN Lin et al. (2019) | 6.95 |

Table 10: FFHQ

| Model | FFHQ 256 × 256 FID |
|---|---|
| UDM (RVE) + ST | 5.54 |
| **Likelihood-free Models** | |
| InsGen Yang et al. (2021) | **3.31** |
| Anycost GAN Lin et al. (2021) | 3.35 |
| StyleGAN2 ADA+bCR Karras et al. (2020) | 3.62 |
| CIPS Anokhin et al. (2020) | 4.38 |
| U-Net GAN Schonfeld et al. (2020) | 7.48 |
| BigGAN Schonfeld et al. (2020) | 11.48 |

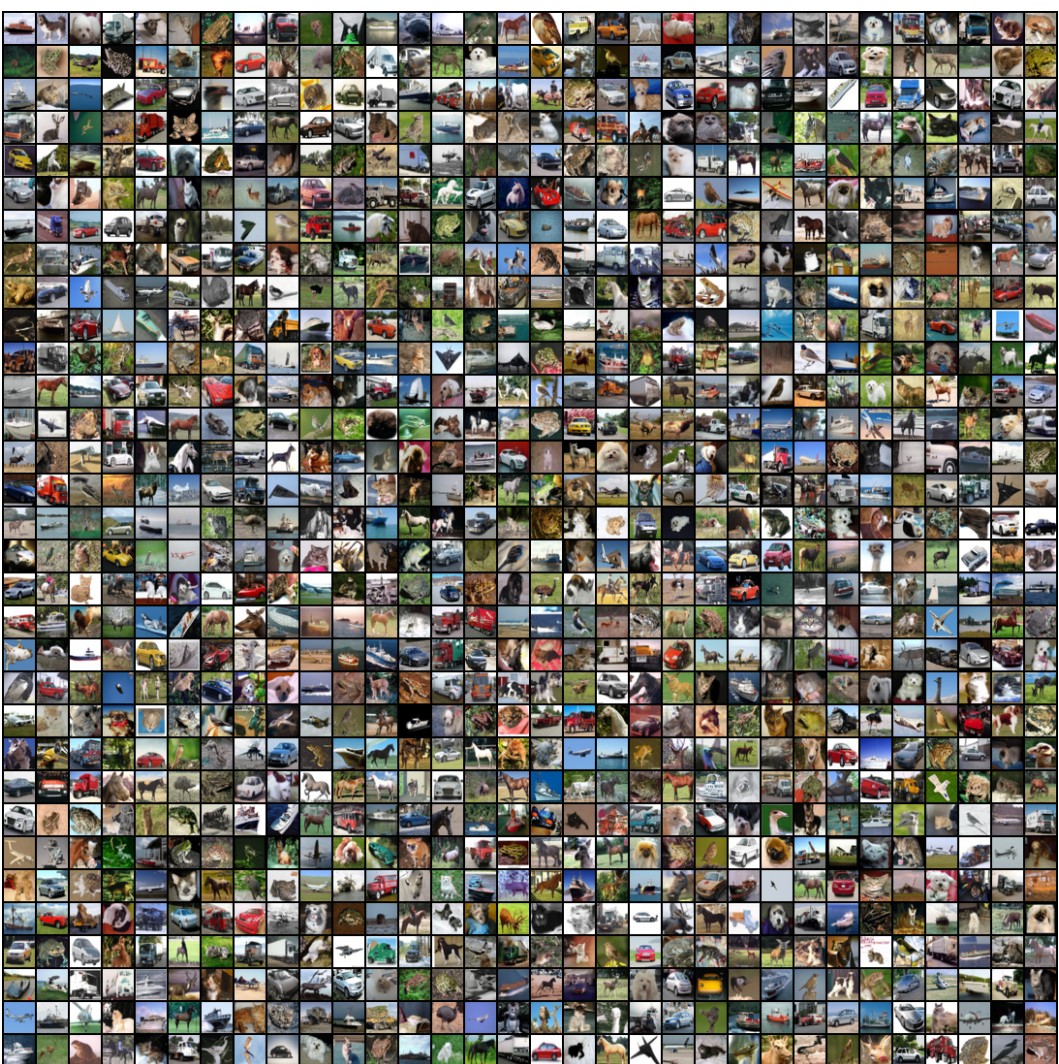

Figure 16: Random samples on CIFAR10.

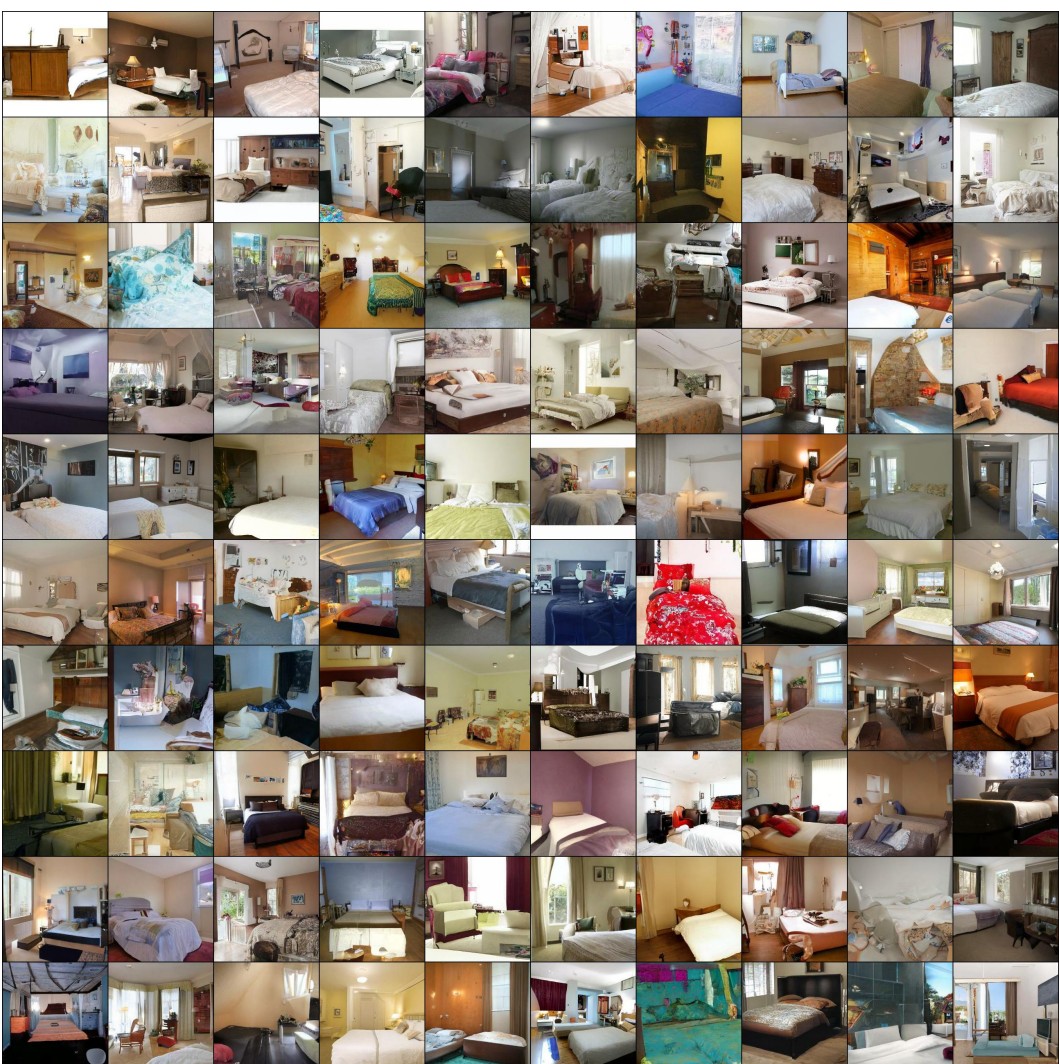

Figure 17: Random samples on LSUN Bedroom.

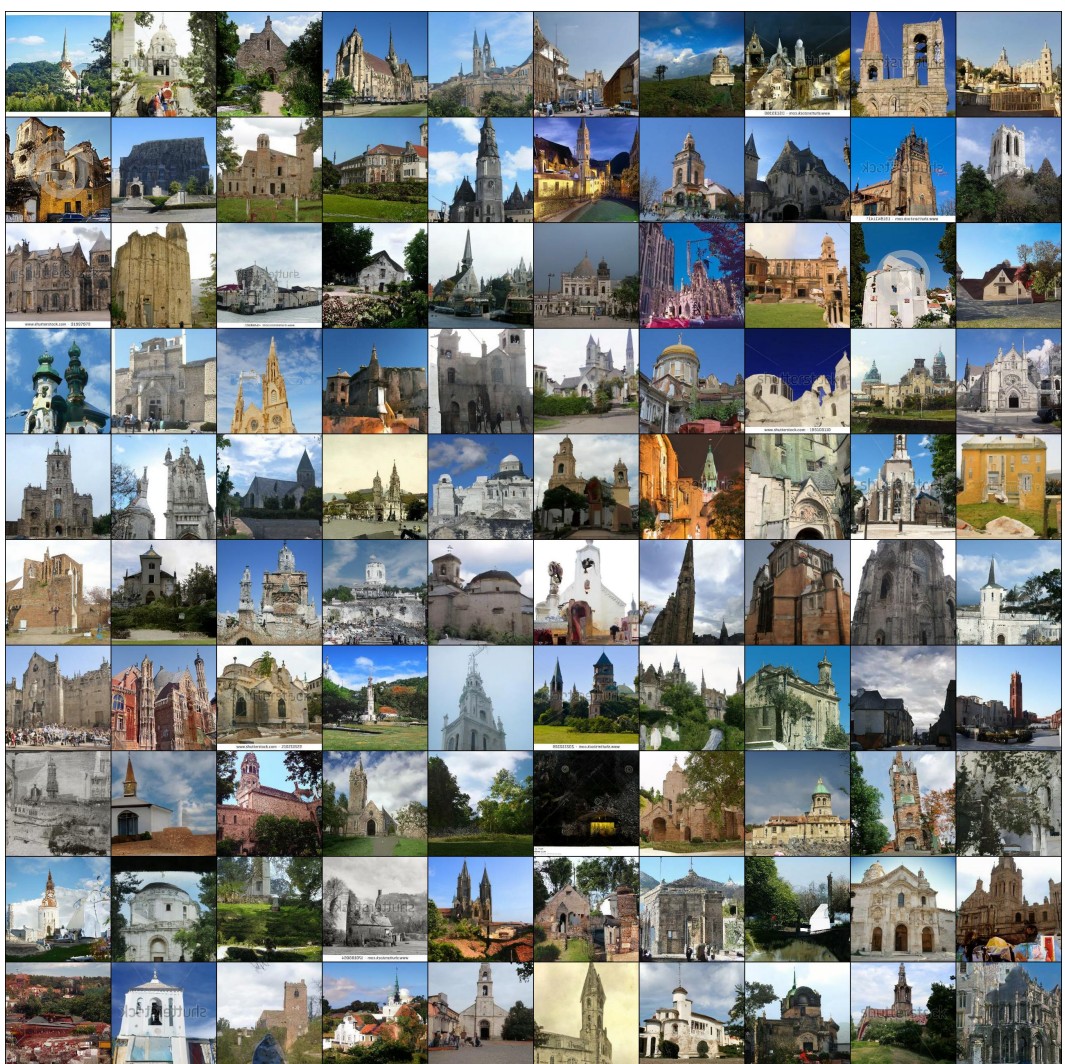

Figure 18: Random samples on LSUN Church.

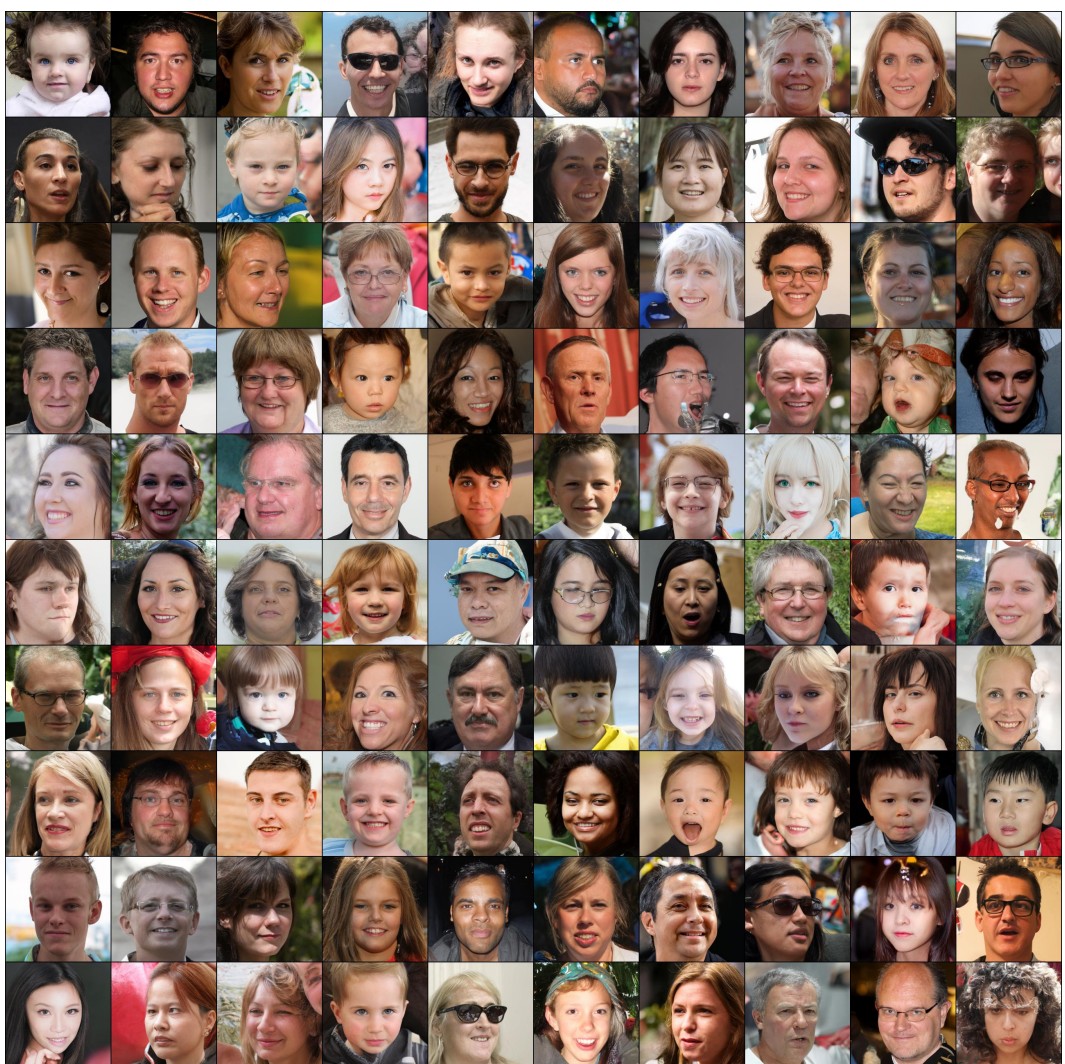

Figure 19: Random samples on FFHQ 256.

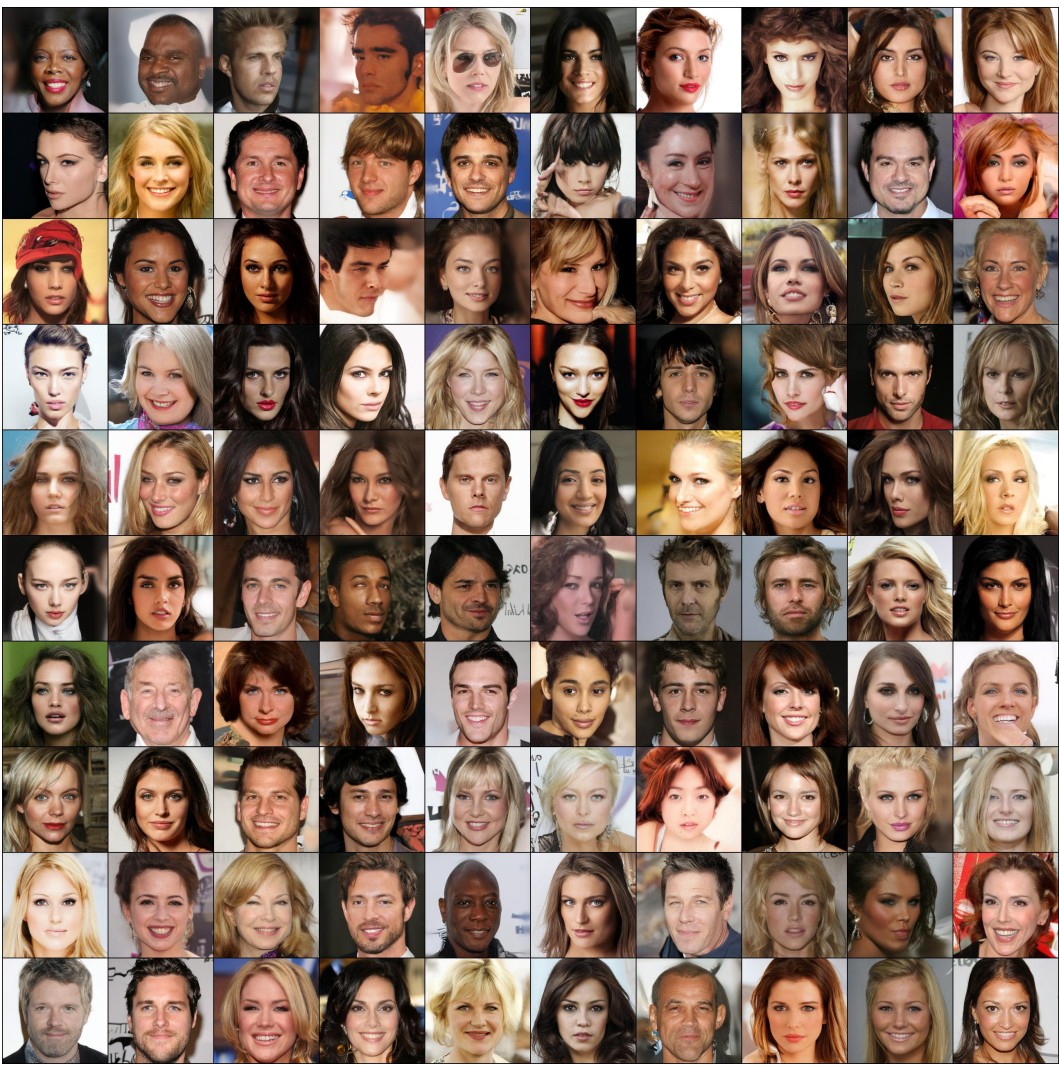

Figure 20: Random samples on CelebA-HQ 256.

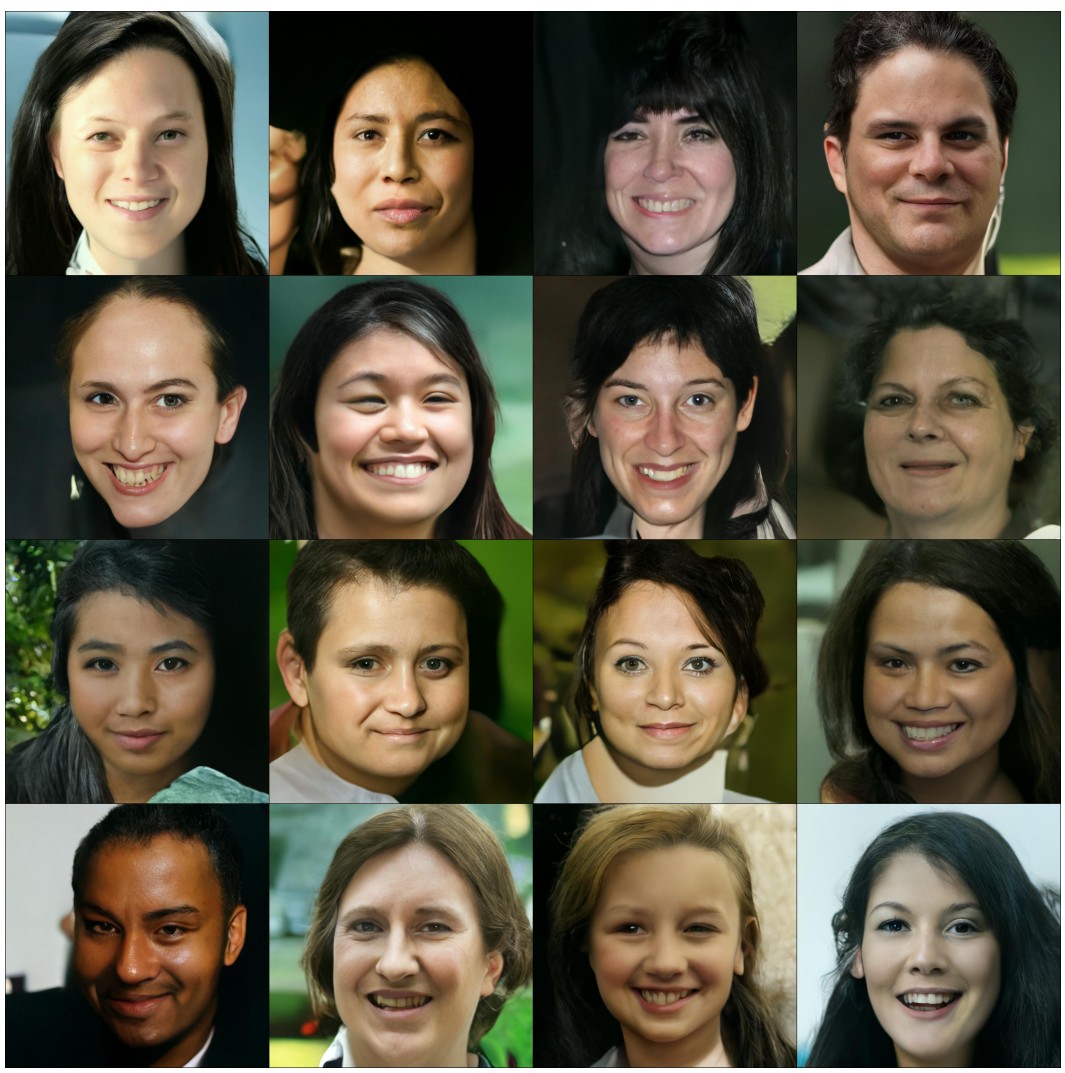

Figure 21: Random samples on FFHQ 1024

