# OpenReview forum: "High Precision Score-based Diffusion Models"
_ICLR.cc/2022/Conference — ICLR 2022 Submitted_

### Official Review · Reviewer_e3pu · 2021-11-01

**Correctness:** 2
**Technical Novelty And Significance:** 2
**Empirical Novelty And Significance:** 2
**Recommendation:** 3
**Confidence:** 4

**Main Review:**

Strengths:

- Overall, the results presented in Table 3 seem quite good. On this note, the improvement on CelebA (64x64) from FID 3.95 of NCSN++ [1] to FID 1.92 (!!) seems really good. I don't assume that the number is wrong, however, I find it suspicious that no samples on this dataset are presented and that the number improved from 2.78 (I am aware of the preprint version of this paper) to 1.92, whereas the NLL number remained the same. I encourage the authors to share checkpoint + minimal FID testing script with the reviewers so that this number can be confirmed.

- (iii) In my opinion, the ST trick is well-motivated and the high-level idea of focusing some batches on smaller time steps and some batches on larger time steps seems intuitive.

- (iv) The authors propose an SDE which leads to simple importance sampling.

- I appreciate that the authors tested their models on several datasets.

Weaknesses:

- The paper uses the probability flow formulation from [1] to compute NLL and the PC sampler from [1] to compute FID and IS. Using different samplers, these are two distinct (!!) models and values of NLL and FID/IS should not be put in the same row as is done in Table 3. This is misleading. I encourage the authors to dedicate one row to FID/IS with PC sample and one row to NLL/FID/IS with ODE sampler. Same comment applies to values reported from other papers in Table 3 and to other Tables in the paper.

- When I was reading the paper (in particular Section 2), I was under the impression that the authors would train their models using likelihood weighting [3] (see, for example, $\lambda = g^2$ in Eq. 1). However, the way Section 4 is written ("On CIFAR-10, we observe that our model improves NCSN++ (VE) in NLL and IS at the expense of sacrificing FID slightly) it seems that the authors used the re-weighted $\lambda$ from [1] for some models. To the best of my knowledge, there is no comment in either main paper or appendix which weighting function $\lambda$ is actually used in the results of tables 3/4/5/7. On the other hand, two different weighting functions are used for Table 6. I encourage the authors to clarify.

- (i) In my opinion, proposing to train models for lower $\varepsilon$ is not so much a contribution but rather just a hyperparameter search. In the paper's defense, NLL & FID do improve for NCSN++ (table 4) when setting $\sigma_{\mathrm{min}}$ from $0.01$ to $0.001$. Decreasing  $\sigma_{\mathrm{min}}$ even further results in insignificant improvements of NLL and worsens FID significantly (Table 6). I think it would have been better to just say that $\sigma_{\mathrm{min}}$ is a hyperparameter and defer the discussion around Figure 4 to the appendix.

- (ii) I disagree with the paper on the significance of this "problem". Generally, score-based generative models are parameterized in such a way that $s_\theta(x, t) = \alpha_\theta(x, t) / r(t)$ (see for example [1]) where $\alpha_\theta$ is a neural network and $r(t) \to 0$ as $t \to 0$. See for example the implementation (https://github.com/yangsong/score_sde_pytorch/blob/1618ddea340f3e4a2ed7852a0694a809775cf8d0/models/utils.py#L129) of [1] which the reviewed paper is based on. So to the best of my knowledge, this parameterization takes care of the "problem". Furthermore, the "performance gain" using the parameterization with $\eta(t)$ in Table 4 (compare row 2 and 3) seems insignificant. Is Figure 13 (a) based on the toy example from Figure 6? If so, I am not sure that the result directly translates to high-dimensional image generation.

- Figure 10: It seems that the FID scores are computed using only 1k generated samples? From my personal experience in computing FID scores, 1000 samples are not enough to make any conclusions (variance is too high). I encourage the authors to repeat the experiment with at least 10k samples.

- (iv) Single ablation (row 3 vs row 4 in Table 4) does not seem enough evidence for justifying RVESDE as an important contribution. I think the authors should have done more experiments, even if only on toy datasets. Furthermore, in my opinion, the main benefit of RVESDE is that importance sampling is simple, however, there is no result shown for RVESDE + importance sampling?!

- The paper is somewhat hard to read and unclear in some claims; see quote from summary.

Suggestions:

- I think the "soft truncation" trick is a cool idea and might solve an important problem. In my opinion, the paper could have been a lot better if this one promising direction would have been studied more in detail rather than studying four problems/solutions.

References:

[3] Song et al. Maximum Likelihood Training of Score-Based Diffusion Models. arXiv:2101.09258v4.

**Summary Of The Paper:**

The paper outlines four problems and proposes a solution for each problem:

(i) Generally, score-based generative models (for example [1]) do not compute the loss on the entire time interval $[0,1]$ but rather only on $[\varepsilon, 1]$, for some $\varepsilon > 0$. The paper claims that choosing $\varepsilon$ too large results in poor details for image generation (see Figure 4), for example, the details (pores) of a human nose.

(ii) The paper shows that the the squared Euclidean norm of the data score (for an unspecified data distribution) empirically diverges as $t \to 0$.  The paper then presents a lemma that states that no neural network can learn an unbounded function as long as it is conditioned on time $t$ or variance $\sigma$. The paper then proposes to condition on $\eta(t)$ where $\eta$ is constructed such that either $\eta(t) \to \infty$ or $\eta(t) \to -\infty$ as $t \to 0$.

(iii) The paper shows that Monte Carlo samples of the likelihood loss have a large variety of scales: values are very large for small times $t$ (or equivalently small noise levels) and very small for large times $t$ (or equivalently large noise levels). Due to the large variety of scales, the authors claim that "the diffusion loss is dominated by the end diffusion time with the front range in practice, and the loss at the end diffusion time is barely counted in the gradient signal. This imbalanced loss brings the immatured score estimation at the end diffusion time, which ruins the overall sample shape." (I am not entirely clear what this quote means.) The paper then proposes a so-called "soft-truncation" (ST) trick. The high-level intuition of the ST trick is described as follows: instead of sampling time points $t$ uniformly over the time range $[\varepsilon, 1]$ in each batch; in each batch it samples a $\tau \sim \mathbb{P}(\tau)$ and then samples $t$ uniformly over the time range $[\tau, 1]$. The authors claim that this has the effect that some batches focus on large time steps (providing a good gradient signal for large noise levels) and some batches focus on small time steps (providing a good gradient signal for small noise levels).  The original setup with fixed $\varepsilon$ can be recovered by setting $\mathbb{P}(\tau)= \delta(\tau = \varepsilon)$. The authors then propose a particular one-parameter ($k$) distribution $\mathbb{P}$ and show different instantiations of $k$. The family is designed in such a way that $\mathbb{P}(\tau) \to \delta(\tau = \varepsilon)$ as $k \to \infty$.

(iv)  The authors show that the continuous VESDE [1],  proposed  as a continuous extension of  [2],  is not actually “geometric” as apparently desired by [2]. The authors then propose the "Reciprocal Variance Exploding" SDE (RVESDE) which can be shown to be "geometric". The authors then claim that importance sampling is simple for the RVESDE.

The paper tests the proposed solutions on CIFAR10 (32x32), CelebA (64x64), CelebA-HQ (256-256), and STL-10 (48x48).

References:

[1] Song et al. Denoising Diffusion Implicit Models. ICLR 2021.

[2] Song & Ermon. Improved Techniques for Training Score-Based Generative Models. NeurIPS 2020.

**Summary Of The Review:**

The paper outlines four problems and one fix for each problem. In my opinion, three fixes are either only incremental or not sufficiently tested. The "soft truncation" trick, on the other hand, seems like a good idea, however, this contribution is buried under the three other problemx/fixes and not enough space/thought is devoted to it. In my opinion, the weaknesses of the paper clearly outperform its strengths.

---

### Official Review · Reviewer_QU8e · 2021-11-03

**Correctness:** 3
**Technical Novelty And Significance:** 3
**Empirical Novelty And Significance:** 4
**Recommendation:** 5
**Confidence:** 3

**Main Review:**

I think paper makes a tangible contribution to the diffusion model literature.  The paper is, however, not very well written. It is quite technical, which makes it challenging to read. But the English is also difficult to understand in places.  I would encourage the authors to address the writing to make it more accessible.  I would further recommend adding more citations throughout, especially in Section 2.

The fact that the score function is unbounded and causes problems for optimization is not particularly surprising.  What is less clear, perhaps, is whether different parameterizations for learning diffusion models suffer from the same root problem.  For example, one might suppose that the epsilon parameterization introduced in the DDPM paper by Ho et al. would not suffer from the same problem. I think this paper would be much stronger if they can claim that other existing parameterizations like that used by Ho et al all suffer from the problem addressed in this paper. This issue should be discussed at length.

Motivation:
The impact of the problems identified close to the beginning and the end of the diffusion process (regarding Figures 3 and 4), are not particularly compelling.  The fact that the hair is synthesized in the wrong region of the image in the early stages of the reverse diffusion process seems to be a relatively weak argument for the problem the authors have identified.  Similarly, regarding Figure 4, it is not entirely clear that the problems that occur as a function of sigma_min being too large are all local, nor is it clear that reducing sigma_min from 10^-2 to 10^-3 has a major impact on image quality in this case.  I am not trying to disagree with the claim that the score function becomes unbounded, and this leads to challenges in the optimization.  Rather, I think the issue is with the impact of the problems raised on the generative model.

The motivation for the ST trick, which appears to be useful, is also not well explained.  More intuitions for it would be very helpful.  That said, this ST-trick appears to be very useful.

Experiments:
The results provided in Table 3 are limited to relatively small images, and in the case of CelebA and CelebA-HQ to datasets with relatively little diversity.  Results are ImageNet 256x256, or even ImageNet 64x64 would also be nice to see, and would also allow comparison with recent SOTA methods like ADM and CDM.

Concerning the quantitative measures of performance, the Appendix states that FID is computed using the InceptionV1 network.  Are the FID numbers with which they compare also computed using InceptiveV1?

The issue of different parameterizations of the loss in diffusion models is also relevant to the experimental work. One important properties of the experiments is the different parameterizations used by different authors in optimizing the diffusion models.  Again, I am thinking of the epsilon parameterization which may not suffer from the same problems as the score function per se. Which methods in Table 3, for example, use which parameterizations? Directly comparing different parameterizations with all other things held constant might be useful for the authors to do directly.

Finally, the paper finds that improved results in training diffusion models occur as sigma_min reduces from 10^-2 to 10^-3, but for values lower than that performance degrades.  This may also be related to conventional quantization levels in images for which there are 256 resolvable values in each color channel at each pixel.  This suggests that adding noise any smaller than 10^-3 would be sufficiently small relative to quantization that there is little signal for denoising at noise levels lower than 10^-3.  To what extent does the lowest effective sigma_min depends on pixel quantization.  For example, with 12 bit medical images, should sigma_min go even lower to further improve performance of the model?

**Summary Of The Paper:**

This paper considers two problems associated with optimizing diffusion models, namely, training instability due to unbounded denoising score functions when the level of noise is very small (at the later stages of the reverse diffusion process), and problems at the earlier stages of the reverse diffusion process where coarse-scale artifacts are sometimes introduced as the corresponding terms of the objective have relatively little influence on the training loss. These problems are addressed through a new parameterization that is better suited to extremely small noise levels, and a soft truncation (ST) trick that better enables the optimization with sufficient influence given to the entire range of noise levels.  The paper has strong technical contributions and very good experimental work, showing improved quantitative measures of performance, namely NLL, FID, and IS.

**Summary Of The Review:**

I think this is an interesting paper.  It is technical and I did not go through all the mathematical details carefully.  However, I think the results are potentially useful and lasting. The paper itself can be cleaned up. The ST-trick appears to be quite useful, but I am less convinced about the significance of the problem with the loss parameterization as discussed above.

---

### Official Review · Reviewer_Veok · 2021-11-03

**Correctness:** 3
**Technical Novelty And Significance:** 2
**Empirical Novelty And Significance:** 4
**Recommendation:** 5
**Confidence:** 4

**Main Review:**

First of all, I would like to applaud the authors for pinpointing the practical issues of training score-based diffusion models and for making an effort in fixing these problems. Below are the strengths of the paper
- The three issues identified in this paper are, to the best of my knowledge, quite relevant in training diffusion models.
- The experiments are well thought-out: the ablation study is quite thorough.

Below are some weaknesses of the paper:
1. Presentation needs more work: the paper contains a lot of imprecise languages and somewhat misleading statements and reasonings. See points (b,c,e,k,l)
2. Questionable technical assumption: The paper hinges heavily on the assumption that the score would blow up (to infinity) as $t\rightarrow0$ (such as Lemma 1, see point (g)). It is not very obvious why this is a reasonable assumption, especially when uniform or variational dequantization is usually performed in practice, even when we consider empirical data distributions that are a mixture of dirac point masses. Perhaps the authors can provide some more explanation in the rebuttal.
3. Uncertainty of the experiments not presented. Are all experiments with a single seed?
4. A lot of grammatical mistakes and typos.

More detailed comments and questions:
a) Intro: I don’t quite understand the logic behind the references in the very first paragraph in the intro. I understand some of them are perhaps state-of-the-arts or perhaps quite new, but I would suggest including a more canonical / standard reference for each category.

b) In section 2, the authors claimed that VESDE and VPSDE are equivalent up to rescaling (and time change). Thus they build their model based on VESDE. But the two SDEs induce very different diffusion, and due to the way neural nets are initialized, this could result in a pretty drastic difference in practice. I’d suggest making this point clearer since in practice training with VESDE does lead to a pretty different performance than VPSDE (as observed in Song et al 2020).

c) Eq (1): the RHS is missing some terms (see Song et al 2021 thm3 or Huang et al 2021 eq (25)). I notice there’s a short note after the score matching loss equality “up to a constant”, but it’s not clear if this applies to eq (1), which can be confusing. Also, I’d suggest perhaps adding the refs there.

d) after eq (2): what is $\sigma(\epsilon)$?

e) Page 3 first paragraph: It says “if the score network exactly estimates the data score, the generated image at time $t$ would follow $x_t$”. This statement feels a bit weird and seems incorrect: score-based diffusion models are not about exactly inverting the perturbing diffusion process. It’s more of a distributional property that the dynamic can be reverted, not a pointwise property.

f) Is Fig 4 a generated sample? Or is it simply a real data point but perturbed with different levels of noise?

g) Lemma 1: the subset U should be open? Also, why is the assumption on an unbounded vector field a reasonable assumption here?

h) Figure 5: How is this curve estimated or computed? Is it for a specific dataset? Is it the score of an empirical distribution?

i) Training instability: from figure 7, I can tell that the loss might have high magnitude and high variance for small sigma. It’s not directly obvious that this will cause training instability. Perhaps a training curve (of avg NLL) will better reflect it? Also, what does different color / different $\tau$ mean?

j) Proposition 1: The VESDE proposed in Song et al. has this problem since a Gaussian noise with variance $\sigma_\min^2$ is always included. That is, they are learning to recover the distribution of $p_\text{date}$ convolved with $N(0, \sigma_\min^2). Not sure it is fair to say VESDE loses its theoretic ground. Also, the same issue does not seem to apply to VPSDE (even though the authors claimed the two are basically equivalent).

k) Section 3.1: Not sure why there is a negative sign in $\eta(t)$ for HNCSN. The optimal score function $s^*$ is a function of $x_t, t$, why is that equal to a random variable $z$? Importance sampling is introduced in the following paragraph without any introduction / reference. $\eta(t)$ for HDDPM is defined as a definite integral? The differential uses the same variable as the argument of $\eta$, which is confusing. What is the $\eta$-distribution in the last paragraph? Please make this paragraph more precise, I don’t understand why this uniformized representation guarantees high precision, what is high precision exactly?  Also, won’t $\eta$ go to infinity as $t$ goes to 0? How come one ends up having a uniform distribution in this unbounded space.

l) Section 3.2: What is the third argument of $\mathcal{L}(\cdot; \cdot, \cdot)$ in the ST-trick? I got the feeling that this is performing random truncation of the perturbing SDE, i.e. the lower integration time ($\epsilon$?) is randomized? Overall the presentation of ST needs more work, I could barely understand what is actually being done here.

m) If the lower integration time is randomized, this still will not result in an unbiased training loss for the lower bound on likelihood right? Then what is the advantage of ST compared to importance sampling proposed in Song et al 2021 and Huang et al 2021?

n) Table 3 is a bit inconclusive, and the improvement of HNCSN++ (RVE) + ST seems a bit marginal. I find it hard to judge the result, what’s the take-home message?

o) In the High Precision Parameterization paragraph of ablation, it says “..with lowered $\sigma_\min effectively reduces the performances”, should it be “improves” instead? The performance gain from the new parameterization is in terms of what? FID?

p) It is unclear for the argument against VESDE using non-trivial $\sigma_\min$, the authors actually tried to sample with $\sigma\rightarrow 0$ after training. This is regarding the paragraph *Optimal Precision* in the ablation. Lowering $\sigma_\min$ will cause numerical instability during training I suppose, but at test time technically one can still extrapolate to smaller $\sigma$. Also, it’s unclear why the 8-bit representation has anything to do with said instability. Isn’t the data uniformly dequantized?


**Summary Of The Paper:**

The paper identifies three problems with score-based diffusion models: 1) unbounded score, 2) training stability, and 3) bias in training using VESDE. The authors then propose the corresponding solution to each problem, with some theoretical motivation, which is then empirically tested.


**Summary Of The Review:**

The paper presents three practical problems and solutions to training / parameterizing score-based diffusion models. Overall the direction is quite relevant, but the presentation of the work is subpar (I’ve pointed at specific sections/paragraphs for future improvement in detailed comments), which makes me doubt it would benefit the ICLR audience *in its current form*.

---

### Official Review · Reviewer_ekSh · 2021-11-03

**Correctness:** 3
**Technical Novelty And Significance:** 2
**Empirical Novelty And Significance:** 2
**Recommendation:** 5
**Confidence:** 3

**Main Review:**

1. For the figures (e.g., figure1,2,5), it would be better to explain in detail how the plots are generated and their settings. For example, in figure 5, it would be important to mention what data distribution it is (is it the distribution in figure 6), are the $y$ axis averaged over one single batch? How many data samples are considered? The result in figure 5 depends a lot on the underlying distribution: if the data distribution is $N(0,I)$,  the $y$ axis (data score) scale might not be that ill-conditioned even when $\sigma$ is small? In many cases, losses with values close to infinity happen at datapoints in low-density regions where scores are not well defined. Thus, a sufficient amount of noise is needed to ensure that the estimated scores are well defined. Thus, in some sense $\sigma_{\text{min}}$ might need to be (strictly) non-zero when the data scores are not well defined.
2. In figure 5, the x, y labels could be improved. Currently the x, y labels do not seem to depend on each other.
3. In figure 6, it seems that the proposed method is still not able to capture the data score well on the toy dataset (although it is slightly better than NCSN++).
4. If we consider $L_t=\mathbb{E}[||s_\theta(x_t,t)-\frac{x-x_t}{\sigma^2}||_2^2]$, and expanding the terms. When $\sigma\to 0$, the entries in the term $(\frac{x-x_t}{\sigma^2})^2$ go to infinity, which could cause $L_t$ to blow up when $\sigma$ is small. When $\sigma\to0$, the training variance can also be large (https://arxiv.org/abs/2002.07501), which causes instability issues. In some sense, tuning the weighting function $g(t)$ might not be able to completely solve the issue that the estimated scores with denoising score matching (DSM) are inaccurate when $\sigma\to0$. To see this, one could use a model conditioned on a single noise scale $\sigma$ (e.g., 1e-3), and plot the accuracy of the estimated score trained with DSM. It seems that the proposed weighting could improve the joint training of conditional models with shared parameters, but might still not be able to solve the large estimation error issue when $\sigma\to0$ (see figure 5). Related to this direction, this paper (https://arxiv.org/abs/2002.07501) proposes a variance reduction technique for DSM with Taylor expansion. This approach often improves the stability and accuracy of training DSM when $\sigma$ is close to zero.
5. The empirical results show that the proposed method allows training the model with a smaller $\sigma_{\text{min}}$ (which could correlate with sample quality) than baseline SDE models. The proposed method also has better sample qualities than current SDE models.



**Summary Of The Paper:**

Diffusion models have achieved great performance in many applications. However, learning a good diffusion model can be challenging since the data score can go to infinity when $t$ goes to zero. To address this issue, this paper proposes an alternative parameterization for unbounded data scores and provides a practical trick (ST-trick) to handle the variation of the scales of score values. It proposes Reciprocal Variance Exploding Stochastic Differential Equation to sample from the model. The proposed method can be applied to existing NCSN and DDPM models. Empirically, it achieves strong performance on high-resolution image generation.

**Summary Of The Review:**

This is a very interesting work that studies the effect of how the sampling steps at different $t$ affect the performance of score-based diffusion models. It also proposes a method to train SDE-based models with smaller $\sigma_\text{min}$ so that samples corresponding to $\sigma_\text{min}$ would ideally have better local fidelity. Empirical results show that the proposed method can improve the sample quality when applied to existing SDE-based models. However, the experimental setup is not very clear and I also have some concerns about the problem formulation (see Main Review).

---

### Official Review · Reviewer_sL3N · 2021-11-05

**Correctness:** 2
**Technical Novelty And Significance:** 3
**Empirical Novelty And Significance:** 3
**Recommendation:** 5
**Confidence:** 3

**Main Review:**

**novelty & significance**

The paper studies a popular model -- the SDE-base score-matching model -- and identifies an important problem. To the best of my knowledge, this exact problem -- issues related to t close to 0 -- has not been studied before in the exact form as is presented in the paper. In this regard, this paper is somewhat novel and its topic suits the venue.

**technical correctness**

I'm quite concerned about this aspect, mostly in terms of the theory and how the theory motivates the narrative of the solutions.

- First, the issue of exploding data scores is only justified through a toy experiment (Figure 5), whose description is quite lacking. For instance, the y-axis in this figure shows that the quantity is $\| \nabla x \log p_t(x_t) \|$. This quantity, however, is a random variable. I am thus confused about what is exactly being shown in the figure -- is it the average data score, is it the max data score over a constrained subset of x-space, or something else? In general, how do you even obtain this data score exactly? Is this a toy example where you know the ground truth data generation model and its density? That's even more surprising, since most densities we use on a day-to-day basis has finite data scores, and thus would suggest a wild discontinuity of the score at time t=0. This seems unlikely to happen if you use smooth forward SDE models.
- Lemma 1 doesn't seem quite right to me. A locally Lipschitz function could be unbounded on a non-compact subset. For instance, consider the function s(x, t) = 1/x defined on (0, 1) x [0, 1]. At t=0, this function could still be unbounded (take x->0). At the same time, it's easy to show that the function is locally Lipschitz, since the derivative is always finite at any input location you pick within the domain (though not bounded anymore).
- The text on the top of page 3 (section about Front Time) seems weird to me as well. The authors have motivated the discussion with the pair of random variables x_0, and x_t where x_t is sampled conditioned on x_0. With this, the authors argue that the L2 distance between a generated sample at small t (denoted x_t' here to distinguish between x_t) and data x_0 can't be close. In this latter conclusion x_0 and x_t' are independent, whereas in the motivating justification, x_0 and x_t follows a Markov chain. The explanation therefore seems like a non-explanation to me, since the assumptions aren't right.

**presentation**

While I can sense that authors are trying to do a good job in this respect, much can still be improved. I find the sections 2.2 and 3 particularly confusing. There are two causes to this confusion:
  - The context of the toy experiment isn't explained carefully. Also, are plots 5, 7, 8 all generated based on this toy experiment? How different quantities in these plots are computed are also left undescribed. This makes evaluating correctness difficult. (recall the issue about data score in the first bullet for the "correctness" section of this review).
  - More motivation and context could be given to certain parts of the main text.

**experiments**

This part seems somewhat insufficient. For instance, authors didn't verify whether the exploding data score issue occurs on natural image data like CIFAR, assuming that Figure 5 is on the toy experiment. Again, I don't quite understand what the y-axis is, since I don't see random fluctuation along the vertical direction of the plot as I expect, since the quantity is a random variable. Ablation studies about the soft-truncation trick and selecting sigma_min seems interesting.

**Summary Of The Paper:**

This paper studies the behaviour of training SDE-based score-matching generative models for time t close to 0. The paper identifies two issues with existing forward SDEs for training such models: 1) the score of the marginal distribution could be large, and 2) the expected squared loss could be large, draining the loss contribution at other times. The paper introduces several remedies to fix this issue.



**Summary Of The Review:**

The paper studies an interesting problem and attempts to present corresponding solutions. The presentation could be improved. Authors should also try to address the issue regarding the correctness of Lemma 1. I am willing to change my mind if the authors could present a convincing argument of why it's true (how my counter argument is wrong), or a fix to the theory.

---

### Decision · Program_Chairs · 2022-01-20

**Decision:**

Reject

**Comment:**

This paper analyzes the behavior of score function as $t \rightarrow$ 0 and proposes simple approaches to mitigate issues around estimating an unbounded data score. The reviewers have acknowledged the importance of the problem and its relevance to current efforts on denoising diffusion models. However, they have also raised serious concerns regarding the clarity of the presentation and missing information across different sections. Additionally, the unbounded data score is only shown on a toy example, and it is not clear if it exists in real-world datasets. Finally, e3pu pointed out that in the commonly used score parameterization from Song et al. the score parameterization can in fact grow to infinity when needed. Given these criticisms and without a response from the authors, we don't believe that the paper is ready for publication at ICLR.